# Numerical homogenization of the viscoplastic behavior of snow based on X-ray tomography images

Antoine Wautier[1,2,3,4,5], Christian Geindreau[1], and Frédéric Flin[2]

[1]Université Grenoble Alpes, CNRS, Grenoble INP, Laboratoire 3SR UMR 5521, F-38000 Grenoble, France.
[2]Météo-France – CNRS, CNRM UMR 3589, CEN, F-38400 Saint Martin d'Hères, France.
[3]Now at AgroParisTech-ENGREF, 19 avenue du Maine, 75732 Paris, France.
[4]Now at Irstea UR RECOVER, 3275 Rte Cézanne, CS 40061, 13182 Aix-en-Provence Cedex 5, France.
[5]Now at Université Grenoble Alpes, Irstea, UR ETGR, 2 rue de la Papeterie-BP 76, F-38402 St-Martin-d'Hères, France.

*Correspondence to:* A. Wautier (antoine.wautier@gmail.com)

**Abstract.** While the homogenization of snow elastic properties has been widely reported in the literature, homogeneous rate dependent behavior responsible for the densification of the snowpack has hardly ever been upscaled from snow microscructure. We therefore adapt homogenization techniques developed within the framework of elasticity to the study of snow visco-plastic behavior. Based on the definition of kinematically uniform boundary conditions, homogenization problems are applied to 3D-images obtained from X-ray tomography, and the mechanical response of snow samples is explored for several densities. We propose an original post-processing approach in terms of viscous dissipated powers in order to formulate snow macroscopic behavior. Then, we show that Abouaf models are able to capture snow visco-plastic behavior and we formulate a homogenized constitutive equation based on a density parametrization. Eventually, we demonstrate the ability of the proposed models to account for the macroscopic mechanical response of snow for classical laboratory tests.

## 1 Introduction

Accurately predicting the macroscopic behavior of snow in a wide range of loads, strain rates and temperatures is of a particular interest with respect to avalanche risk forecasting or to structural design of avalanche defense structures. The macroscopic mechanical behavior of snow is however strongly influenced by many microscale parameters such as its density (Mellor, 1974), its microstructure topology (Shapiro et al., 1997) or the mechanical behavior of its ice matrix (elastic, visco-plastic, brittle-failure). Disregarding the influence of temperature (Schweizer and Camponovo, 2002), snow exhibits two very different types of mechanical behavior depending on the strain rate (Schulson et al., 2009). At high strain rate, large deformations are mainly controlled by bond failures and grain rearrangements, whereas at very low strain rates (typically smaller than $10^{-5}\text{s}^{-1}$), snow exhibits a visco-plastic behavior (Narita, 1984; Salm, 1982) which plays an important role in the long-term densification of the snowpack through the microscale deformation of the ice skeleton.

During the last decades, many experiments have been performed to explore the macroscopic behavior of different snow types under various loading conditions and temperatures (Mellor, 1974; Salm, 1982; Desrues et al., 1980; Shapiro et al., 1997;

Bartelt and von Moos, 2000; Moos et al., 2003; Scopozza and Bartelt, 2003a). Within the framework of continuum mechanics, several phenomenological models have then been proposed in order to account for these experimental data (Desrues et al., 1980; Scopozza and Bartelt, 2003b; Cresseri and Jommi, 2005; Navarre et al., 2007; Cresseri et al., 2009). The fitted material parameters arising in these models often only characterize the mean properties of a few types of snow in a restricted density range. Thanks to the recent application of X-ray tomography to snow (Brzoska et al., 1999; Schneebeli, 2004; Kaempfer et al., 2005; Flin and Brzoska, 2008; Chen and Baker, 2010; Srivastava et al., 2010; Pinzer et al., 2012; Wang and Baker, 2013; Adams and Walters, 2014; Calonne et al., 2015) good databases of 3D images for the main snow types reported in the international classification (Fierz et al., 2009) are now available (Calonne et al., 2012; Löwe et al., 2013). Given these extensive geometrical descriptions of snow, its corresponding macroscopic behavior can be up-scaled in a more systematic way thanks to the use of techniques derived from the homogenization theory (Dormieux and Bourgeois, 2002; Auriault et al., 2010).

In recent numerical studies, discrete element (DEM) and finite element (FEM) methods are the main techniques used to bridge the gap between the topology of the ice skeleton and snow macroscopic mechanical behavior.

- In DEM, the snow skeleton is viewed as an assembly of ice grains interacting with each other at contact points. This method is well suited to model complex interfacial interactions between snow grains, possibly leading to grain rearrangements such as elasto-viscoplastic contact deformation, grain sintering, bond breakage or grain sliding. It has been already used on 3D idealized assemblies of ice grains to identify microstructural deformation mechanisms and to simulate creep densification processes (Johnson and Hopkins, 2005). Application of DEM directly on X-ray tomography images is however not straightforward, since the ice skeleton must be segmented into ice grains approximated with simple geometrical shapes. Recently, DEM simulations taking into account cohesion and friction at the contact between grains have been performed on 3D assemblies of grains deduced from X-ray tomography and approximated by clumps of spheres (Hagenmuller et al., 2015). In addition, at the scale of ice grains, the crystalline orientation is known to play a role on the viscous deformation mechanisms at the contact between two grains (secondary creep), as it has been recently reported by Burr et al. (2015b, a, 2017). The full granular structure of snow (grain shape and crystal orientations) can be determined on real 3D images of snow by X-ray Diffraction Contrast Tomography (DCT) (Rolland du Roscoat et al., 2011) but the application of this technique is not straightforward and very few images are available.

- In FEM, the complex 3D snow skeleton observed by X-ray microtomography can be meshed without loosing any information on the microstructure and different types of mechanical behavior can be assumed for polycrystalline ice. While grain rearrangements cannot be considered without the use of mesh adaptation techniques, the deformation of the ice skeleton is explicitly taken into account, which is of primary importance as long as the topology of the ice skeleton is preserved. In the last decade, many studies addressed in details the case of elasticity (Schneebeli, 2004; Pieritz et al., 2004; Srivastava et al., 2010; Köchle and Schneebeli, 2014; Wautier et al., 2015; Srivastava et al., 2016), possibly up to a brittle failure (Hagenmuller et al., 2014) whereas more complex types of constitutive behavior were hardly ever considered. To the best of our knowledge, no general visco-plastic constitutive equation for snow can be found in the literature.

Such an equation would however be of particular interest to address the long term densification of the snowpack under its own weight. During this process, snow deforms at a low strain rate and an important part of the densification results from the viscous deformation (secondary creep) of the ice skeleton. Among the scarce microscale FEM modeling of snow plasticity, Theile et al. (2011) proposed a beam network model based on 3D images to simulate uniaxial creep of snow, Chandel et al. (2014) used an elasto-plastic constitutive law for ice in order to determine failure envelopes.

In the following, we formulate a 3D macroscopic viscoplastic constitutive law for snow by performing FEM simulations on 3D images. As performed in Wautier et al. (2015) within the framework of elasticity, typical kinematically uniform boundary conditions (KUBC) are applied to 3D images of snow, and finite element simulations are run in order to link the macroscopic stress response of different snow samples to the imposed strain rates in an incremental form. For each numerical simulation, only small strains are considered to avoid any important microstructure modification. Then, the macroscopic law is generalized to finite deformation problems thanks to the use of a collection of 3D snow images exhibiting different microstructures and densities. In this upscaling process, the viscous behavior of ice is described by a power law of exponent $n$ (secondary creep) as in Theile et al. (2011). Due to the non-linear behavior under consideration, the homogenization does not provide the complete structure of the macroscopic constitutive equation (Auriault et al., 1992, 2002; Geindreau and Auriault, 1999; Orgéas et al., 2007). Nevertheless, it can be shown that the exponent $n$ is preserved at the macroscopic scale and that the macroscopic dissipation power is the volume-averaged of the local one (Suquet, 1993). Using these properties, the macroscopic constitutive equation of snow is formulated within the framework defined by the theory of representation of anisotropic tensor functions (Smith, 1971; Liu, 1982) and by using macroscopic isodissipation surfaces (Green, 1972; Abouaf, 1985; Duva and Crow, 1992; Sofronis and McMeeking, 1992; Geindreau et al., 1999b; Storakers et al., 1999; Sanchez et al., 2002; Orgéas et al., 2007).

The paper is organized as follows. In section 2, the numerical homogenization procedure used in Wautier et al. (2015) is recalled and adapted to the study of non-linear constitutive equations. In section 3, the post-processing procedure used in order to characterize the macroscopic viscous behavior of snow in terms of macroscopic isodissipation curves is presented. These curves might be seen as the equivalent of yield surfaces in plasticity as they characterize the set of stress or strain rate states leading to the same level of mechanical dissipation. Their shapes result from the strong coupling between the microstructure of the ice skeleton and the ice viscous behavior at the microscale. They characterize the 3D viscoplastic behavior of snow. In section 4, we show that Abouaf models (Abouaf, 1985) are well suited to describe the macroscopic viscous behavior of snow deduced from our numerical simulations. In the end of this section, we propose a macroscopic formulation of the viscoplastic behavior of snow. Finally, in section 5, the mechanical responses of snow for classical experimental tests (uniaxial, oedometric and triaxial compression tests) are modeled thanks to our upscaled law. This illustrates the potential applications of our 3D homogenized constitutive behavior.

## 2 Numerical homogenization procedure: from image to macroscopic mechanical response

Based on the homogenization theory, it is often possible to replace a heterogeneous material by an equivalent homogeneous one provided that its microstructure is sufficiently small with respect to the macroscopic scale of interest. With respect to snow, this separation of scale hypothesis is satisfied in most of the cases and its macroscopic mechanical behavior can be deduced from mesovolumes obtained thanks to X-ray tomography. Previous studies showed that in most of the cases, samples of a few millimeters can be considered as representative elementary volumes (REV) for the study of the mechanical behavior of snow (Wautier et al., 2015; Srivastava et al., 2016). In the following, in order to distinguish the two scales of interest, lowercase letters are used for microscopic quantities while uppercase ones are used for their macroscopic counterparts.

Irrespective of the size of the sample considered, the boundary conditions used in a homogenization procedure introduce undesired boundary effects of varying thickness. Depending on the type of boundary conditions used, the size of the REV should be adapted accordingly. Three particular types of boundary conditions are considered to give relatively small REV. In decreasing order of REV (Kanit et al., 2003), these are statically uniform boundary conditions (SUBC), with a macroscopic homogeneous stress imposed on the boundary, kinematically uniform boundary conditions (KUBC), with a macroscopic homogeneous strain imposed on the boundary, and periodic boundary conditions (PBC), with a periodicity condition imposed on the displacement field and the normal stress across the sample boundaries. Although PBC are considered to give the best convergence with respect to the size of the REV (Kanit et al., 2003), their application to a non-periodic highly porous microstructure is not straightforward. It is necessary, for example, to enclose the sample by a virtual boundary or to assume that the pores are filled by a soft material. In order to avoid the introduction of such artifacts, KUBC were retained. The KUBC numerical homogenization procedure introduced in Wautier et al. (2015) is used in this paper and easily adapted to the study of the elasto-viscoplastic behavior of snow. It consists in the four steps recalled in Figure 1.

The first two steps remain unchanged and consist in: (i) meshing the 3D-microtomographic images (Step 1), (ii) defining the kinematic relation $\boldsymbol{u} = \mathbf{E} \cdot \boldsymbol{x}$ between the homogeneous macroscopic strain $\mathbf{E}$ and the displacement field $\boldsymbol{u}$ on the boundary (Step 2). Step 1 requires the use of the MATLAB open-source toolbox iso2mesh (Fang and Boas, 2009) while step 2 is achieved thanks to the use of the plug-in Homtools (Lejeunes et al., 2011). More details can be found in Wautier et al. (2015). The next two steps (Step 3 and Step 4) are modified in order to take into account the change in the constitutive modeling of ice.

### 2.1 Elasto-viscoplastic behavior of ice (Step 3)

In the following, the mechanical behavior of the polycrystalline ice is supposed to be elasto-viscoplastic and isotropic. The total strain rate tensor ($\dot{\boldsymbol{\varepsilon}}$) is decomposed as the sum of an elastic part ($\dot{\boldsymbol{\varepsilon}}_e$) and a viscous part ($\dot{\boldsymbol{\varepsilon}}_v$) as

$$\dot{\boldsymbol{\varepsilon}} = \dot{\boldsymbol{\varepsilon}}_e + \dot{\boldsymbol{\varepsilon}}_v. \tag{1}$$

The elastic part can be expressed as, $\boldsymbol{\varepsilon}_e = (\mathbf{C}^{\text{ice}})^{-1} : \boldsymbol{\sigma}$, where $\mathbf{C}^{\text{ice}}$ is the elastic stiffness tensor, $\boldsymbol{\sigma}$ is the Cauchy stress tensor and ":" the double contraction product. Due to isotropy, $\mathbf{C}^{\text{ice}}$ is fully defined by a Young's modulus $E$ and a Poisson ratio $\nu$.

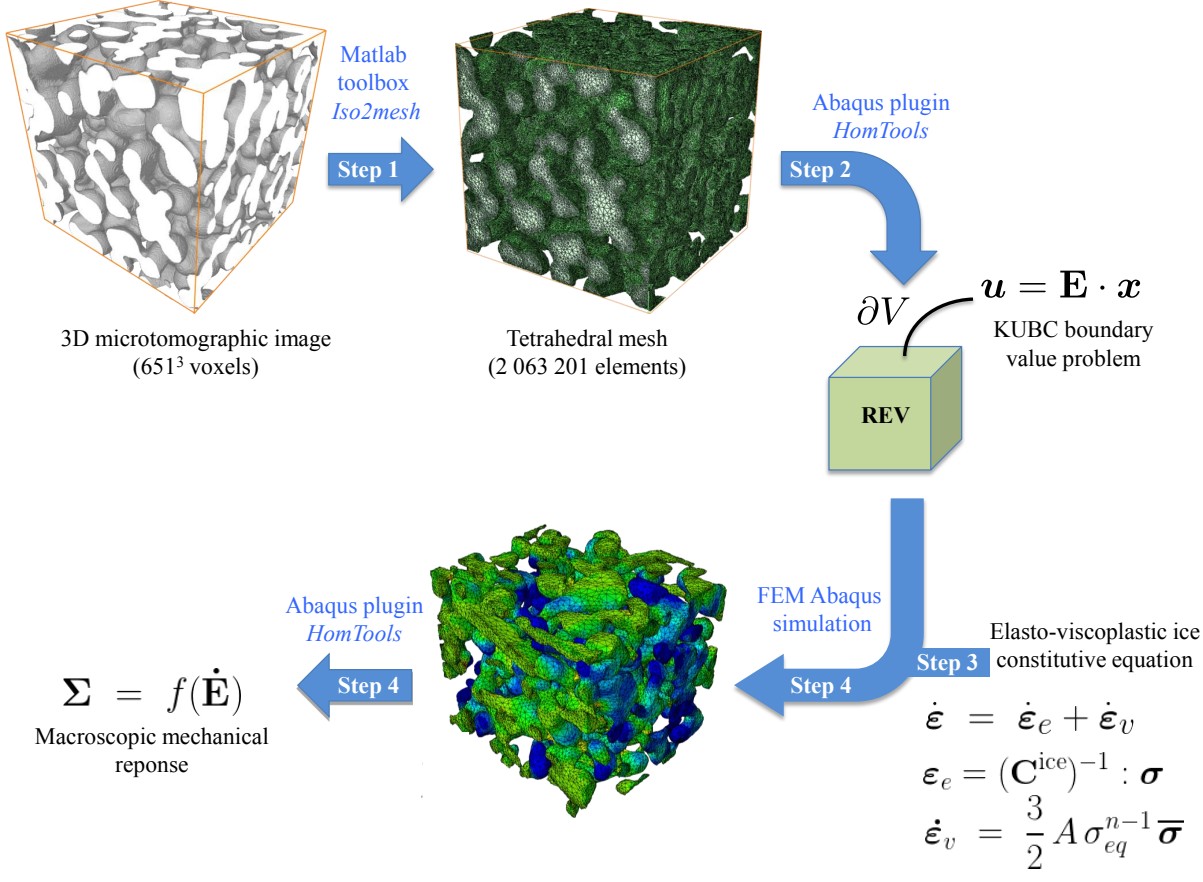

**Figure 1.** Four-step procedure used in order to transform 3-D microtomograph images of snow into finite element models and numerically solve KUBC homogenization boundary value problems.

Concerning the viscous part, at low strain rates, the non linear mechanical behavior of ice is usually described by a power law (Mellor, 1974; Schulson et al., 2009), i.e. the Norton Hoff in 3D (Lemaitre and Chaboche, 1985). Thus, we have

$$\dot{\boldsymbol{\varepsilon}}_v = \frac{3}{2} A \sigma_{\mathrm{eq}}^{(n-1)} \overline{\boldsymbol{\sigma}}, \tag{2}$$

where $A$ and $n$ are two material parameters (which usually depend on the temperature), $\overline{\boldsymbol{\sigma}}$ is the deviatoric stress tensor and $\sigma_{\mathrm{eq}}(\boldsymbol{\sigma})$ is the equivalent stress defined as

$$\sigma_{\mathrm{eq}}(\boldsymbol{\sigma}) = \sqrt{\frac{3}{2}\overline{\boldsymbol{\sigma}}:\overline{\boldsymbol{\sigma}}} = \sqrt{\frac{3}{2}}\bar{s}_2, \qquad \overline{\boldsymbol{\sigma}} = \boldsymbol{\sigma} - \frac{1}{3}\mathrm{Tr}(\boldsymbol{\sigma})\mathbf{I}, \tag{3}$$

where $\mathbf{I}$ is the second order identity tensor, Tr is the trace operator and $\bar{s}_2$ is the second invariant of $\overline{\boldsymbol{\sigma}}$. It can be shown (Lemaitre and Chaboche, 1985) that the viscous strain rate tensor $\dot{\boldsymbol{\varepsilon}}_v$ derives from a viscous potential $\omega(\boldsymbol{\sigma})$ as

$$\dot{\boldsymbol{\varepsilon}}_v = \frac{\partial \omega}{\partial \boldsymbol{\sigma}} = \frac{\mathrm{d}\omega}{\mathrm{d}\sigma_{\mathrm{eq}}}\frac{\partial \sigma_{\mathrm{eq}}}{\partial \boldsymbol{\sigma}} = \frac{3}{2}\frac{\mathrm{d}\omega}{\mathrm{d}\sigma_{\mathrm{eq}}}\frac{\overline{\boldsymbol{\sigma}}}{\sigma_{\mathrm{eq}}}, \qquad \text{with} \quad \omega(\boldsymbol{\sigma}) = \frac{A}{n+1}\sigma_{\mathrm{eq}}^{(n+1)}. \tag{4}$$

From (2) and (4), one can define the equivalent strain rate $\dot{\varepsilon}_{\mathrm{eq}}(\dot{\varepsilon}_v)$ as the dual variable of the equivalent stress $\sigma_{\mathrm{eq}}$ such that

$$\dot{\varepsilon}_{\mathrm{eq}}(\dot{\varepsilon}_v) = \sqrt{\frac{2}{3}\dot{\bar{\varepsilon}}_v : \dot{\bar{\varepsilon}}_v} = \sqrt{\frac{2}{3}}\bar{e}_2, \quad \text{with} \quad \dot{\varepsilon}_{\mathrm{eq}} = A\sigma_{\mathrm{eq}}^n, \tag{5}$$

where $\bar{e}_2$ is the second invariant of the deviatoric part of $\dot{\varepsilon}_v$. If $p_{\mathrm{v}}$ stands for the volumetric mechanical dissipation, the equivalent stress and the equivalent strain rate verify

$$p_{\mathrm{v}} = \boldsymbol{\sigma} : \dot{\bar{\varepsilon}}_v = \bar{s}_2.\bar{e}_2 = \dot{\varepsilon}_{\mathrm{eq}}.\sigma_{\mathrm{eq}} = A\sigma_{\mathrm{eq}}^{(n+1)}. \tag{6}$$

At the microscopic scale, the viscoplastic deformation of ice is incompressible. Consequently the equivalent stress (resp. the equivalent strain rate) depends only on the second invariant $\bar{s}_2$ of $\overline{\boldsymbol{\sigma}}$ (resp. the second invariant $\bar{e}_2$ of $\dot{\bar{\varepsilon}}_v$).

Overall, the ice matrix is thus modeled as an isotropic elasto-viscoplastic material in the finite element commercial software Abaqus. The values of the material constants used in this constitutive modeling, namely $A$, $n$, $E$ and $\nu$, are given in Table 1. It should be underlined that even if the most common values used for $E$ and $n$ are $E = 9$ GPa and $n = 3$, the ice Young's modulus values found in the literature range from 0.2 to 9.5 GPa (Chandel et al., 2014) and the values for $n$ vary between 1.8 and 4.6 under usual loading and temperature conditions (Scapozza and Bartelt, 2003; Schulson et al., 2009; Schleef et al., 2014). As a result, different values for $E$ and $n$ are considered in this study to give more insight on their influence on the homogenized viscoplastic behavior of snow.

**Table 1.** Mechanical parameters used in the elasto-viscoplastic modeling of ice implemented in Abaqus.

| Parameters | Value |
|---|---|
| $A$ | $1.5\ 10^{-3}\,\mathrm{MPa}^{-n}.\mathrm{s}^{-1}$ |
| $n$ | 2, 3, 4.5 |
| $E$ | 325 MPa, 9 GPa |
| $\nu$ | 0.3 |

## 2.2 Macro-strain paths definition (Step 4)

Given a time dependent macroscopic strain loading $\mathbf{E}(t)$, the KUBC homogenization problem to be solved reads as

$$
\begin{cases}
\operatorname{div}\boldsymbol{\sigma} & = & 0 & \text{for } \boldsymbol{x} \in V \\[4pt]
\boldsymbol{u} & = & \mathbf{E}(t) \cdot \boldsymbol{x} & \text{for } \boldsymbol{x} \in \partial V \\[4pt]
\boldsymbol{\varepsilon} & = & \dfrac{1}{2}\left(\boldsymbol{\nabla}\,\boldsymbol{u} + {}^{t}\boldsymbol{\nabla}\,\boldsymbol{u}\right) & \text{for } \boldsymbol{x} \in V \\[4pt]
\boldsymbol{\varepsilon} & = & \boldsymbol{\varepsilon}_e + \boldsymbol{\varepsilon}_v & \text{for } \boldsymbol{x} \in V \\[4pt]
\dot{\boldsymbol{\varepsilon}}_v & = & \dfrac{3}{2}\,A(\boldsymbol{x})\,\sigma_{\mathrm{eq}}^{n-1}\,\overline{\boldsymbol{\sigma}} & \text{for } \boldsymbol{x} \in V \\[4pt]
\boldsymbol{\sigma} & = & \dfrac{E(\boldsymbol{x})}{1+\nu}\left(\boldsymbol{\varepsilon}_e + \dfrac{\nu}{1-2\nu}\operatorname{Tr}(\boldsymbol{\varepsilon}_e)\mathbf{1}\right) & \text{for } \boldsymbol{x} \in V
\end{cases}
\tag{7}
$$

where $V$ stands for the domain occupied by the whole snow sample and $\partial V$ its boundary. The spatial heterogeneity of the mechanical properties of snow is captured thanks to the functions $A(\boldsymbol{x})$ and $E(\boldsymbol{x})$ defined as

$$
A(\boldsymbol{x}) = \begin{cases} A & \text{if } \boldsymbol{x} \in V_i \\ 0 & \text{if } \boldsymbol{x} \in V \backslash V_i \end{cases}
\quad \text{and} \quad
E(\boldsymbol{x}) = \begin{cases} E & \text{if } \boldsymbol{x} \in V_i \\ 0 & \text{if } \boldsymbol{x} \in V \backslash V_i \end{cases}.
\tag{8}
$$

where $V_i \subset V$ is the domain occupied by the ice matrix. Similarly to the elastic case (Wautier et al., 2015), the macroscopic stress tensor $\boldsymbol{\Sigma}$ is deduced from the knowledge of its microscopic counterpart thanks to the volume averaging

$$
\boldsymbol{\Sigma} = \frac{1}{|V|}\int_V \boldsymbol{\sigma}\,\mathrm{d}V = \langle \boldsymbol{\sigma} \rangle
\tag{9}
$$

As a result, for a given macroscopic strain loading $\mathbf{E}(t)$, the macroscopic stress response $\boldsymbol{\Sigma}(t)$ is recovered. The implicit function linking these two second order tensors characterizes the homogeneous behavior of the snow sample considered and can be put in the form:

$$
\dot{\mathbf{E}} = \dot{\mathbf{E}}_e + \dot{\mathbf{E}}_v = \mathcal{F}(\boldsymbol{\Sigma})
\tag{10}
$$

where $\dot{\mathbf{E}}_e$ is the macroscopic elastic strain rate tensor and $\dot{\mathbf{E}}_v$ is the macroscopic viscous strain rate tensor. The elastic part can be expressed as, $\mathbf{E}_e = (\mathbf{C}^{\mathrm{hom}})^{-1} : \boldsymbol{\Sigma}$, where $\mathbf{C}^{\mathrm{hom}}$ is the homogenized stiffness tensor (Wautier et al., 2015). This tensor can be obtained by performing only six simulations on Representative Elementary Volumes extracted from 3D images. A single simulation is required if the snow microstructure is isotropic. By contrast, the homogenization of the visco-plastic behavior requires *a priori* an infinite number of numerical simulations. However, this number of simulations can be reduced by taking into account some theoretical results (Auriault et al., 1992; Suquet, 1993; Orgéas et al., 2007). Indeed, it can be shown that:

– The homogeneity of degree $n$ of the microscopic viscous constitutive equation (2) is preserved in the homogenization process. In other words, the macroscopic viscous strain rate $\dot{\mathbf{E}}_v$ is a homogeneous function of degree $n$ of the macroscopic

stress $\mathbf{\Sigma}$, and the macroscopic volumetric mechanical dissipation $\mathcal{P}_{\mathrm{v}} = \dot{\mathbf{E}}_v : \mathbf{\Sigma}$ is an homogeneous function of degree $n+1$ of $\mathbf{\Sigma}$

$$\begin{cases} \dot{\mathbf{E}}_v(\lambda\mathbf{\Sigma}) &= \lambda^n\,\dot{\mathbf{E}}_v(\mathbf{\Sigma}) \\ \mathcal{P}_{\mathrm{v}}(\lambda\mathbf{\Sigma}) &= \lambda\mathbf{\Sigma}:\dot{\mathbf{E}}_v(\lambda\mathbf{\Sigma}) = \lambda^{n+1}\,\mathcal{P}_{\mathrm{v}}(\mathbf{\Sigma}) \end{cases}, \qquad \forall\lambda\in\mathbb{R}. \tag{11}$$

As a result, the choice in the macroscopic strain rate $\dot{\mathbf{E}}_v$ can be reduced to the unit sphere in the second order tensor space, i.e. to strain rate tensors of norm $\sqrt{\dot{\mathbf{E}}_v : \dot{\mathbf{E}}_v} = 1$.

– The macroscopic dissipation potential $\Omega(\mathbf{\Sigma})$ is the volume-average of the local dissipation potential $\omega$

$$\Omega(\mathbf{\Sigma}) = \frac{1}{|V|}\int_V \omega(\boldsymbol{\sigma})\,\mathrm{d}V = \langle\omega(\boldsymbol{\sigma})\rangle \tag{12}$$

and consequently, as at the microscopic scale (see equation (4)), we have

$$\dot{\mathbf{E}}_v = \frac{\partial\Omega}{\partial\mathbf{\Sigma}} = \frac{\mathrm{d}\Omega}{\mathrm{d}\Sigma_{\mathrm{eq}}}\frac{\partial\Sigma_{\mathrm{eq}}}{\partial\mathbf{\Sigma}}, \qquad \text{with} \quad \Omega(\mathbf{\Sigma}) = \frac{A}{n+1}\Sigma_{\mathrm{eq}}^{n+1} \tag{13}$$

where $\Sigma_{\mathrm{eq}}(\mathbf{\Sigma})$ is the macroscopic equivalent stress.

$\Sigma_{\mathrm{eq}}(\mathbf{\Sigma})$ verifies

$$\mathcal{P}_{\mathrm{v}} = \mathbf{\Sigma}:\dot{\mathbf{E}}_v = \dot{\mathrm{E}}_{\mathrm{eq}}.\Sigma_{\mathrm{eq}} = A\Sigma_{\mathrm{eq}}^{n+1} \tag{14}$$

with $\dot{\mathrm{E}}_{\mathrm{eq}}(\dot{\mathbf{E}}_v)$ the macroscopic equivalent strain rate defined by duality.

As a result, the macroscopic viscoplastic law (13) is perfectly defined if the macroscopic equivalent stress $\Sigma_{\mathrm{eq}}$ is known. The latter equation (14) shows that this macroscopic equivalent stress $\Sigma_{\mathrm{eq}}$ can be fitted on iso-mechanical dissipation surfaces in the space associated with $\mathbf{\Sigma}$. Let us remark that the shape and size of such iso-mechanical dissipation surfaces result from the strong coupling between the microstructure and the non-linear behavior of the ice under consideration. The relation (14) also shows the equivalent stress can be obtained whatever the chosen $A$ value. In the case of general anisotropy, the form of $\Sigma_{\mathrm{eq}}$ can be formulated within the framework defined by the theory of representation of anisotropic tensor functions (Smith, 1971; Liu, 1982). It is also important to mention that for the ice matrix, the overall response of snow is insensitive to the sign of $\mathbf{\Sigma}$ as a consequence of definition (4). This condition may be expressed as $\Omega(\mathbf{\Sigma}) = \Omega(-\mathbf{\Sigma})$. Finally, let us remark that by definition (see (13) and (14)), the macroscopic strain rate $\dot{\mathbf{E}}_v$ is normal to iso-mechanical dissipation surfaces (normality rule).

In the following, for the sake of simplicity, we will suppose that the macroscopic viscoplastic behavior of snow is isotropic. In this particular case, for a given value of $n$, it can be shown (Abouaf, 1985; Geindreau et al., 1999b; Danas et al., 2008) that the macroscopic equivalent stress is written:

$$\Sigma_{\mathrm{eq}}(\mathbf{\Sigma}) = \Sigma_{\mathrm{eq}}(S_1, \bar{S}_2, \bar{S}_3, \phi) \tag{15}$$

where $\phi$ is the snow porosity and $(S_1, \bar{S}_2, \bar{S}_3)$ are the three invariants of the macroscopic stress tensor $\boldsymbol{\Sigma}$ defined as:

$$S_1 = \text{Tr}(\boldsymbol{\Sigma}), \quad \bar{S}_2 = \sqrt{\overline{\boldsymbol{\Sigma}} : \overline{\boldsymbol{\Sigma}}}, \quad \bar{S}_3 = \det(\overline{\boldsymbol{\Sigma}}), \quad \text{with} \quad \overline{\boldsymbol{\Sigma}} = \boldsymbol{\Sigma} - \frac{S_1}{3}\mathbf{I}. \tag{16}$$

Similarly, the macroscopic equivalent strain rate $\dot{\text{E}}_{\text{eq}}$ takes the form

$$\dot{\text{E}}_{\text{eq}}(\dot{\mathbf{E}}_v) = \dot{\text{E}}_{\text{eq}}(E_1, \bar{E}_2, \bar{E}_3, \phi) \tag{17}$$

where $(E_1, \bar{E}_2, \bar{E}_3)$ are the invariants of the strain rate tensor $\dot{\mathbf{E}}_v$ defined as

$$E_1 = \text{Tr}(\dot{\mathbf{E}}_v), \quad \bar{E}_2 = \sqrt{\dot{\overline{\mathbf{E}}}_v : \dot{\overline{\mathbf{E}}}_v}, \quad \bar{E}_3 = \det(\dot{\overline{\mathbf{E}}}_v), \quad \text{with} \quad \dot{\overline{\mathbf{E}}}_v = \dot{\mathbf{E}}_v - \frac{E_1}{3}\mathbf{I}. \tag{18}$$

In contrast with the microscopic scale (see equation (3)), the macroscopic equivalent stress (15) depends on the three invariants $(S_1, \bar{S}_2, \bar{S}_3)$ of the macroscopic stress tensor $\boldsymbol{\Sigma}$. Indeed, at the macroscopic scale, the viscoplastic snow deformation is compressible. This compressibility, characterized by $E_1$, depends on the level of the mean pressure $(S_1/3)$ applied on the snow sample, as well as the mean shear stress $(\bar{S}_2)$. The third invariant $\bar{S}_3$ characterizes the loading type and is linked to the Lode angle $\theta$ in the stress space (Lemaitre and Chaboche, 1985; Danas et al., 2008)

$$\cos(3\theta) = \frac{27}{2} \frac{\bar{S}_3}{\Sigma_{\text{eq}}^3}. \tag{19}$$

As a first approximation, it seems reasonable to assume that the influence of the third invariant $\bar{S}_3$ is negligible (Green, 1972; Abouaf, 1985; Geindreau et al., 1999b; Fritzen et al., 2012). Consequently, the macroscopic volumetric mechanical dissipation $\mathcal{P}_{\text{v}}$ depends on the first and second stress and strain invariants and not only on the second ones as at the microscale (6).

$$\mathcal{P}_{\text{v}} = \boldsymbol{\Sigma} : \dot{\mathbf{E}}_v = \text{E}_{\text{eq}}.\Sigma_{\text{eq}} = A\Sigma_{\text{eq}}^{n+1} = \frac{1}{3}E_1.S_1 + \bar{E}_2.\bar{S}_2. \tag{20}$$

The relation (20) shows that, for a given snow porosity, the equivalent macroscopic stress $\Sigma_{\text{eq}}$ can be fitted on iso-volumetric mechanical dissipation curves in the plane $(S_1/3, \bar{S}_2)$. These isodissipation curves can be obtained by plotting the values $(S_1/3, \bar{S}_2)$ corresponding to different loading conditions defined by $(E_1, \bar{E}_2)$. Therefore, the choice was made to run numerical simulations for seven diagonal strain rate tensors defined such that the loading direction in the plane $(E_1, \bar{E}_2)$ varies from $0°$ to $90°$. More explicitly, $\dot{\mathbf{E}}$ applied on the sample is taken as

$$\dot{\mathbf{E}} = \dot{\text{E}}_{\text{ref}} \begin{pmatrix} 1 & 0 & 0 \\ 0 & \eta & 0 \\ 0 & 0 & \eta \end{pmatrix}, \tag{21}$$

with $\dot{\text{E}}_{\text{ref}} = 10^{-7} \text{ s}^{-1}$ and $\eta \leq 1$ such that $\dfrac{\bar{E}_2}{E_1} = \sqrt{\dfrac{2}{3}} \dfrac{1-\eta}{1+2\eta} = \tan\theta, \quad \theta \in \{0°, 9°, 18°, 30°, 45°, 65°, 90°\}$.

Finally, to be consistent with the isotropy hypothesis, numerical simulations have been performed on the most isotropic snow samples with respect to their elastic behavior from the snow database used in Wautier et al. (2015). With reference to the

supporting information of the cited paper (Wautier et al., 2015), the name and the principal characteristics of each sample are recalled in Table 2. The porosities of the selected samples vary from 0.43 to 0.87, which covers almost the entire range of porosity of seasonal snow. Each sample presents similar correlation lengths $(\ell_1, \ell_2, \ell_3)$ (Löwe et al., 2013; Calonne et al., 2014) in the three space directions. All the simulations have been performed on volumes extracted from the 3D images sufficiently large to be considered as REV, as in Wautier et al. (2015).

**Table 2.** Names and principal characteristics of the six snow images of Wautier et al. (2015) used in this study.

| Sample name | Snow type | Dim (px) | Dim (mm) | Resolution ($\mu$m/px) | Snow density (kg/m$^3$) | Porosity | Correlation lengths $(\ell_1, \ell_2, \ell_3)$ ($\mu$m) |
|---|---|---|---|---|---|---|---|
| PP_123kg_600 | PP | 600 | 2.95 | 4.91 | 123.31 | 0.87 | (64, 64, 65) |
| RG_172kg_600 | RG | 600 | 2.95 | 4.91 | 172.74 | 0.81 | (92, 94, 97) |
| RG_256kg_512 | RG | 512 | 2.51 | 4.91 | 256.28 | 0.72 | (113, 111, 110) |
| RG_1600 | RG | 600 | 4.46 | 7.43 | 330.13 | 0.64 | (117, 111, 108) |
| RG_430kg_651 | RG | 651 | 5.61 | 8.61 | 430.59 | 0.53 | (83, 82, 81) |
| MF_522kg_542 | MF | 542 | 5.42 | 10.00 | 522.31 | 0.43 | (138, 134, 133) |

## 3 Post-processing procedure: from macroscopic stress response to a homogenized model for snow visco-plasticity

From the homogenization procedure presented in the previous section, the time response of a given isotropic snow sample is recovered for the seven loading directions in the plane of the strain invariants $(E_1, \bar{E}_2)$ given by the equation (21). The strain rate is applied on each sample during less than 40,000 s, corresponding to a volumetric strain smaller than 1.2 %. The overall viscous behavior of the snow samples is deduced thanks to a post-processing procedure consisting in the three steps described in this section (steps a to c) and summarized in Figure 2.

### 3.1 Extracting the viscous response (step a)

Because snow is locally modeled as an elasto-visco-plastic material in Abaqus (see subsection 2.1), the macroscopic time response $\boldsymbol{\Sigma}(t) = \frac{1}{|V|} \int_V \boldsymbol{\sigma}(\boldsymbol{x}) \, \mathrm{d}V$ deduced from numerical simulations does not involve only viscoplasticity. In the case where the ice viscosity is activated everywhere in the ice skeleton, the macroscopic stress $\boldsymbol{\Sigma}(t)$ corresponding to a constant strain rate should stabilize around a constant value according to equation (2). However, due to the complex geometry of the ice skeleton, the ice viscosity is not uniformly activated and the time response of $\boldsymbol{\Sigma}(t)$ is influenced by the ice elastic behavior even in the long term. Based on the material parameters $A$, $n$, and $E$, and the typical imposed strain rate $\dot{\mathrm{E}}_{\mathrm{ref}}$, a characteristic time $\tau$ can be introduced as the ratio between the ice viscosity $\eta(\dot{\mathrm{E}}_{\mathrm{ref}}) = (\dot{\mathrm{E}}_{\mathrm{ref}}/A)^{1/n}/\dot{\mathrm{E}}_{\mathrm{ref}}$ and the Young modulus $E$

$$\tau = \frac{\eta(\dot{\mathrm{E}}_{\mathrm{ref}})}{E} = \frac{1}{E} \left( A^{-\frac{1}{n}} \dot{\mathrm{E}}_{\mathrm{ref}}^{\frac{1-n}{n}} \right). \tag{22}$$

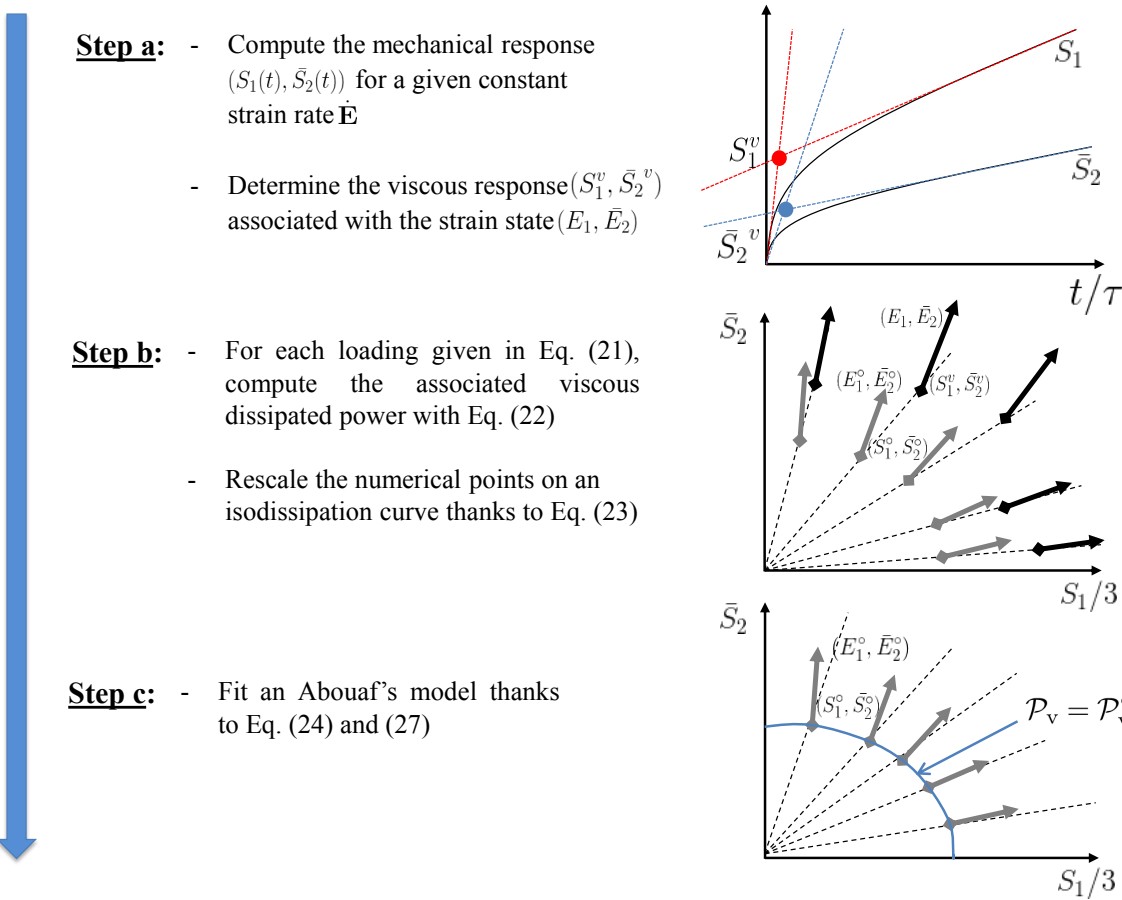

**Step a:** - Compute the mechanical response $(S_1(t), \bar{S}_2(t))$ for a given constant strain rate $\dot{\mathbf{E}}$

         - Determine the viscous response $(S_1^v, \bar{S}_2^{\ v})$ associated with the strain state $(E_1, \bar{E}_2)$

**Step b:** - For each loading given in Eq. (21), compute the associated viscous dissipated power with Eq. (22)

         - Rescale the numerical points on an isodissipation curve thanks to Eq. (23)

**Step c:** - Fit an Abouaf's model thanks to Eq. (24) and (27)

**Figure 2.** Three-step post-processing procedure used in order to formulate a homogenized viscous constitutive equation.

The typical stress response of a snow sample under a constant given strain rate versus the dimensionless time $(t/\tau)$ is illustrated in Figure 3 for the snow sample *RG_1600* (Table 2) for different values of $n$ and $E$. In all the cases, the mechanical response is characterized by a transient regime driven by the elastic properties followed by a permanent regime dominated by the viscoplastic behavior. As illustrated by the comparison between the cases $(n, E) = (4.5, 325 \text{ MPa})$ and $(n, E) = (4.5, 9 \text{ GPa})$, the responses $S_1(t/\tau)$ and $\bar{S}_2(t/\tau)$ are independent of the Young's modulus value chosen. On the contrary, the mechanical response is influenced by the $n$ value.

As a result, for a given value of $n$ and whatever the Young's modulus, the viscoplastic behavior of snow can be characterized by computing the intersection point between initial and final asymptotes of the curves $S_1(t/\tau)$ and $\bar{S}_2(t/\tau)$ in Figure 3. The obtained values for the two stress invariants are noted $S_1^v$ and $\bar{S}_2^v$ and are systematically used in the rest of this paper as the snow viscous homogeneous response to a given imposed constant macroscopic strain rate. The top graph in Figure 2 illustrates this step of the post-processing procedure.

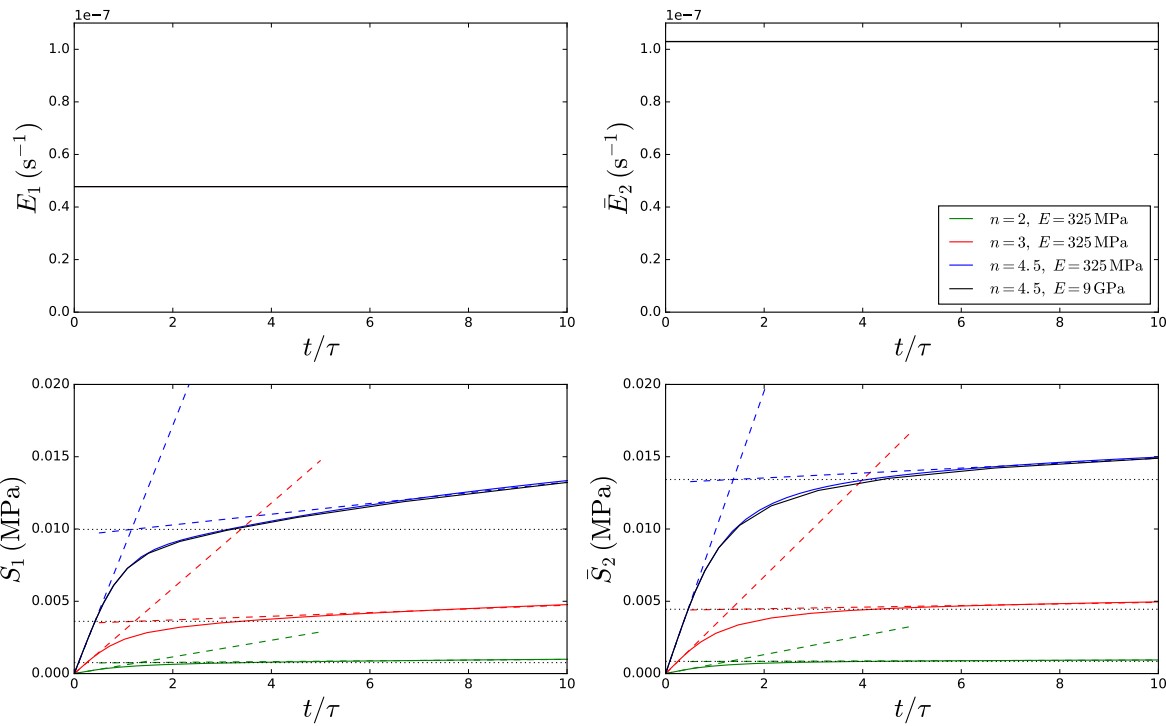

**Figure 3.** Imposed strain rate (top) and stress response (bottom) versus dimensionless time ($t/\tau$) for the sample *RG_1600* (see Table 2). The loading strain rate is characterized by $\theta = 65°$ in equation (21). Two Young's moduli and three values of $n$ are considered.

### 3.2 Computing isodissipation curves (step b)

For a given snow sample of porosity $\phi$ and for each applied loading path (21), the macroscopic volumetric mechanical dissipation $\mathcal{P}_\mathrm{v}$ (20) is computed as

$$\mathcal{P}_\mathrm{v} = \frac{1}{3} E_1 . S_1^v + \bar{E}_2 . \bar{S}_2^v, \tag{23}$$

5    where $S_1^v$ and $\bar{S}_2^v$ are the characteristic stress invariants obtained in the first step of the post-processing process (Figure 2). Each loading path leads to different values of $\mathcal{P}_\mathrm{v}$. However, iso-mechanical dissipation points in the plane $(S_1/3, \bar{S}_2)$ can be recovered thanks to the homogeneity property (11). Given an arbitrary value of $\mathcal{P}_\mathrm{v}^\circ = 1\,\mathrm{Pa.s}^{-1}$, the corresponding macroscopic strain and stress invariants are computed as

$$\left( E_1^\circ, \bar{E}_2^\circ \right) = \left( \frac{\mathcal{P}_\mathrm{v}^\circ}{\mathcal{P}_\mathrm{v}} \right)^{n/(n+1)} \left( E_1, \bar{E}_2 \right), \quad \text{and} \quad \left( S_1^\circ, \bar{S}_2^\circ \right) = \left( \frac{\mathcal{P}_\mathrm{v}^\circ}{\mathcal{P}_\mathrm{v}} \right)^{1/(n+1)} \left( S_1^v, \bar{S}_2^v \right). \tag{24}$$

Thanks to this rescaling, the seven homogenization tests (21) enable the description of an isodissipation curve in the plane of the stress invariants $(S_1/3, \bar{S}_2)$ as illustrated in Figure 2 (step b). For each point $(S_1^\circ/3, \bar{S}_2^\circ)$ on this graph, the associated flow vector $(E_1^\circ, \bar{E}_2^\circ)$ is plotted. The viscous dissipated power is thus simply equivalent to the scalar product $(S_1^\circ/3, \bar{S}_2^\circ) \cdot (E_1^\circ, \bar{E}_2^\circ)$.

### 3.3   Abouaf's model (step c)

Within the framework presented in section 2.2, Abouaf (1985) has suggested to use the macroscopic equivalent stress initially proposed by Green (1972) to describe the viscoplastic behavior of metal powders at high temperatures. This macroscopic equivalent stress $\Sigma_{\text{eq}}(\boldsymbol{\Sigma})$, is written

$$\Sigma_{\text{eq}}(\boldsymbol{\Sigma}) = \Sigma_{\text{eq}}(S_1, \bar{S}_2, \phi) = \sqrt{f(\phi) S_1^2 + \frac{3}{2} c(\phi) \bar{S}_2^2}, \tag{25}$$

where $f(\phi)$ and $c(\phi)$ are two material functions which depend on snow porosity only for a given exponent $n$ of the constitutive equation (2). When $\phi = 0$, we have $f(\phi) = 0$ and $c(\phi) = 1$ in order to recover the equivalent viscous stress of the ice matrix (3): $\Sigma_{\text{eq}}(\boldsymbol{\Sigma}, \phi = 0) = \sigma_{\text{eq}}(\boldsymbol{\sigma})$. From the definition of the viscous strain in (13) together with the previous definition of the equivalent stress in (25), it can be shown that

$$\dot{\mathbf{E}}_v = A \Sigma_{\text{eq}}^{n-1} \left( f(\phi) S_1 \mathbf{I} + \frac{3}{2} c(\phi) \overline{\boldsymbol{\Sigma}} \right). \tag{26}$$

As a result, the corresponding macroscopic equivalent strain rate introduced in equation (20) reads

$$\dot{\mathbf{E}}_{\text{eq}}(\dot{\mathbf{E}}_v) = \dot{\mathbf{E}}_{\text{eq}}(E_1, \bar{E}_2, \phi) = \sqrt{\frac{E_1^2}{9 f(\phi)} + \frac{2}{3} \frac{\bar{E}_2^2}{c(\phi)}}. \tag{27}$$

For a given porosity $\phi$, the combination of (25) and (20) provides an implicit definition of $f(\phi)$ and $c(\phi)$ such that, for all $(S_1, \bar{S}_2)$

$$\Sigma_{\text{eq}}(S_1, \bar{S}_2, \phi) = \sqrt{f(\phi) S_1^2 + \frac{3}{2} c(\phi) \bar{S}_2^2} = \left( \frac{\mathcal{P}_v^\circ}{A} \right)^{1/(n+1)}. \tag{28}$$

In the present work, optimal values for $f(\phi)$ and $c(\phi)$ were obtained in the range $\phi \in [0.43, 0.87]$, by minimizing the quadratic error between the model (28) and the numerical points $(S_1^\circ(\theta)/3, \bar{S}_2^\circ(\theta))$.

## 4   Results and discussion

The homogenization and the post-processing procedure presented in the previous sections are applied to six isotropic snow samples of various densities chosen in the same database as Wautier et al. (2015) and already introduced in Table 2. In Figure 4, the seven points $(S_1^\circ/3, \bar{S}_2^\circ)$ corresponding to the strain rates of equation (21) are represented for these six snow samples in the plane of the two first stress invariants for $n = 4.5$. Similar results have been obtained for the other values of $n$, as shown on the Figure 5. The corresponding strain flow vectors $(E_1^\circ, \bar{E}_2^\circ)$ are shown by solid arrows and isodissipation curves corresponding to fitted Abouaf models are represented by solid lines. Optimal values for $f$ and $c$ obtained for each snow type are presented on Figure 4 and reported in Table 3 for $n \in \{2, 3, 4.5\}$. It should be underlined here that each isodissipation curve is typical of a given snow characterized by its density and thus each curve is also an iso-density curve.

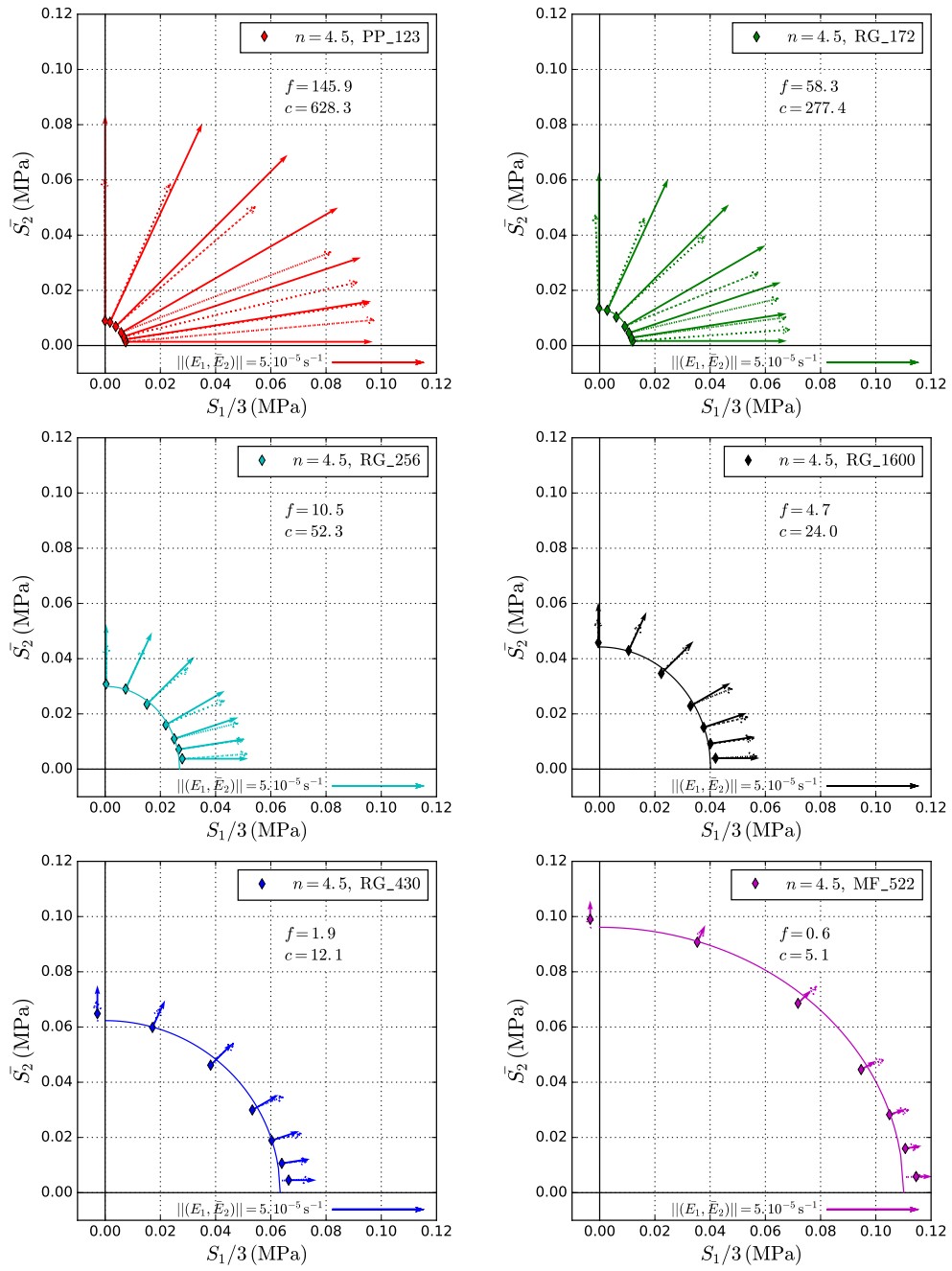

**Figure 4.** Isodissipation curves in the plane of the stress invariants $(S_1, \bar{S}_2)$ corresponding to $n = 4.5$ and to an arbitrary dissipated power $\mathcal{P}_v^\circ$ for the six snow samples of Table 2 in increasing density order. The associated strain flow vectors $(E_1^\circ, \bar{E}_2^\circ)$ are represented by solid arrows. Abouaf models are fitted to the numerical points (solid lines) and theoretical values of strain flow vectors are shown (dashed arrows). Parameters $f$ and $c$ of the fits are shown on the graphs and summarized in Table 3.

## 4.1 Isodissipation curves for various snow samples

The comparison between simulated points and fitted Abouaf models shows overall good agreement even if a systematic slight deviation is observed for the highest $S_1$ values. It should also be noticed that the stress state corresponding to an isotropic strain rate ($\bar{E}_2 = 0$) is not completely isotropic ($\bar{S}_2 \neq 0$). This feature cannot be captured by Abouaf models, which assumes a perfect isotropic behavior of the material. Even if the snow samples used in this study were selected as isotropic as possible, a slight anisotropy should account for the observed residual deviatoric stress component existing under an isotropic strain loading.

As already mentioned in section 2.2, the viscous behavior of snow should be insensitive to the sign of $\mathbf{\Sigma}$ as the ice matrix behave exactly the same in tension and in compression. In the stress space $(S_1/3, \bar{S}_2)$, this results in the symmetry of the isodissipation curves with respect to the axis $S_1/3 = 0$. Provided that the isodissipation curves are smooth, their tangent for $S_1/3 = 0$ is horizontal, which is fulfilled in Figure 4. It must be mentioned that when snow is subjected to large strain levels, geometrical effects will introduce non linear effects and the mechanical response in tension will differ from the one in compression. These effects can also be investigated using the same homogenization procedure.

The overall viscoplastic response of snow is of course sensitive to the $n$ exponent of the Norton Hoff's law (2) used for the ice (see section 3.1). As for example, in Figure 5, the influence of $n$ onto the isodissipation curves is shown for the snow sample *MF_522*. Similar results have been obtained on the other samples. As expected, for a given value $\mathcal{P}_v^\circ$, the size of the isodissipation curves increases with $n$ (since the ice viscosity $\eta(\dot{E}_{ref})$ increases) but their shape remains unchanged. They can be deduced from each other by simple dilation.

## 4.2 Density dependence of the isodissipation curves

As snow density increases, isodissipation curves tend to expand, and conversely flow vectors tend to get smaller. In terms of physics, this means that the denser the snow, the smaller the applied strain rate in order to dissipate the same level of viscous power. In the meantime the applied stress should be increased. This is consistent with the fact that fresh snow tends to get denser more rapidly than already compacted snow under the same imposed loading.

**Table 3.** Optimal values for the parameters $f$ and $c$ of the Abouaf's equivalent stress (25) for three $n$ values.

| Sample name | Porosity | $n = 2$ | | $n = 3$ | | $n = 4.5$ | |
|---|---|---|---|---|---|---|---|
| | | $f$ | $c$ | $f$ | $c$ | $f$ | $c$ |
| PP_123kg_600 | 0.87 | 36.0 | 150 | 79.7 | 336 | 146 | 628 |
| RG_172kg_600 | 0.81 | 16.3 | 75.7 | 33.5 | 156 | 58.3 | 277.4 |
| RG_256kg_512 | 0.72 | 4.05 | 20.5 | 6.98 | 34.7 | 10.5 | 52.3 |
| RG_1600 | 0.64 | 2.07 | 11.0 | 3.32 | 17.0 | 4.70 | 24.0 |
| RG_430kg_651 | 0.53 | 0.915 | 6.38 | 1.40 | 9.12 | 1.89 | 12.1 |
| MF_522kg_542 | 0.43 | 0.354 | 3.32 | 0.503 | 4.26 | 0.630 | 5.07 |

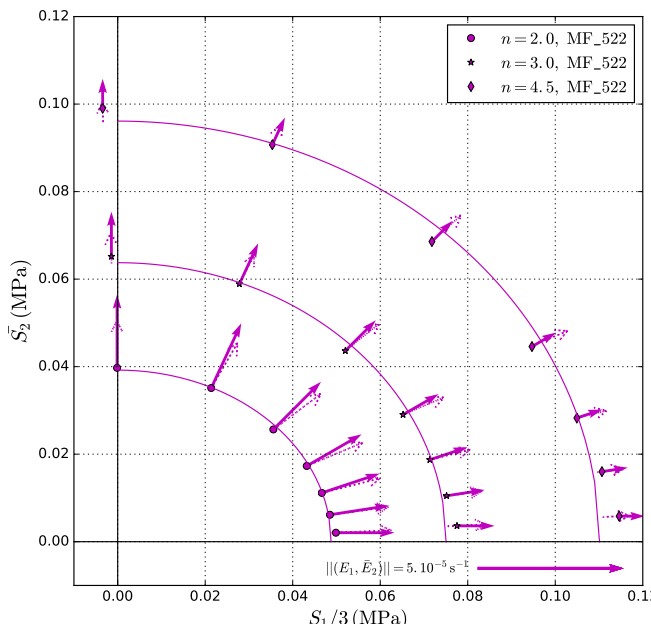

**Figure 5.** Influence of the exponent $n$ onto the isodissipation curves ($\mathcal{P}_v = \mathcal{P}_v^\circ$) for the particular snow sample *MF_522*. The associated strain flow vectors $(E_1^\circ, \bar{E}_2^\circ)$ are represented by solid arrows. Abouaf models are fitted to the numerical points (solid lines) and theoretical values of strain flow vectors are shown (dashed arrows).

The density dependence of snow viscous behavior is fully described by the evolution of $f(\phi)$ and $c(\phi)$ with respect to snow compacity ($\rho_{\mathrm{snow}}/\rho_{\mathrm{ice}} = 1 - \phi$) represented in Figure 6 for $n \in \{2, 3, 4.5\}$. As snow density increases in Figure 6, parameter values decrease, which is consistent with the implicit definition of the isodissipation curves in equation (25). For a given equivalent stress $\Sigma_{\mathrm{eq}}$, higher values for $f$ and $c$ will result in lower stress invariants $S_1$ and $\bar{S}_2$ as observed in Figure 4.

5 Concerning the influence of $n$, the observed increase in $f$ and $c$ is consistent with the dilation of the isodissipation curves observed in Figure 5.

Different expressions of the material functions $f(\phi)$ and $c(\phi)$ have been proposed in the past based on experimental data on metal powders (Abouaf, 1985; Abouaf and Chenot, 1988; Geindreau et al., 1999b), micromechanical modeling (cell model - Duva and Crow (1992)) or numerical simulations on simple microstructures (Sofronis and McMeeking, 1992). These functions

10 have been identified in a restricted range of porosity (dense materials with $\phi < 0.4$). We propose to fit our numerical results using the expressions proposed by Geindreau et al. (1999b). In order to account for the change in the porosity range between metal powders and snow, the compacity limit value of 0.57 is set equal to zero. As a result, the proposed functions are written

$$\begin{cases} f(\phi) = a\left(\dfrac{\phi}{1-\phi}\right)^p \\ c(\phi) = 1 + b\left(\dfrac{\phi}{1-\phi}\right)^q \end{cases}, \quad (a,b,p,q) \in \mathbb{R}^4. \tag{29}$$

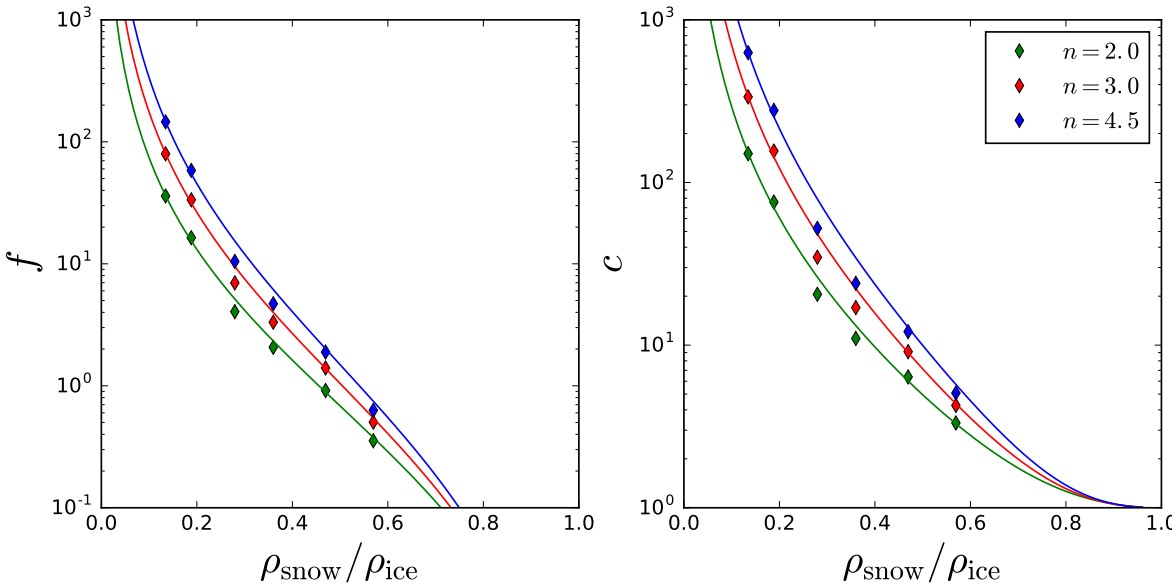

**Figure 6.** Evolution of the Abouaf coefficients $f$ and $c$ (numerical results as diamond points, functions (29) as solid lines) with respect to snow compacity for different $n$ values.

**Table 4.** Optimal parameters chosen for the expressions (29) for different $n$ values.

|   | $n=2$ | $n=3$ | $n=4.5$ |
|---|-------|-------|---------|
| $a$ | 0.68 | 1.0 | 1.5 |
| $p$ | 2.1 | 2.3 | 2.5 |
| $b$ | 4.0 | 6.1 | 8.9 |
| $q$ | 2.0 | 2.2 | 2.3 |

The above fits respect the theoretical values $f(0) = 0$ and $c(0) = 1$ already mentioned in section 3.3. For highly porous snow ($\phi \to 1$), an infinitely small stress level would be needed in order to produce a high viscous dissipation. This is consistent with the infinitely high values for $f$ and $c$ proposed by the above functions (29). These functions allow a good description of the numerical points resulting from the homogenization of the six snow samples (Table 3) and are represented by solid lines in
5   Figure 6 for $n \in \{2, 3, 4.5\}$. As a result, they may stand for a general formulation for the viscous isotropic behavior of snow according to its porosity through the four $n$-dependent parameters $(a, b, p, q)$ given in Table 4. For the sake of illustration, the evolution of the coefficients $a$, $p$, $b$ and $q$ with respect to $n$ is shown in Figure 7. Let us remark that the parameters $a$ and $b$ are close to the ones obtained for metal powders by Geindreau et al. (1999b). However, the exponents corresponding to the snow case are approximately twice bigger, which is linked to a more pronounced dependence on the porosity for very porous
10  materials.

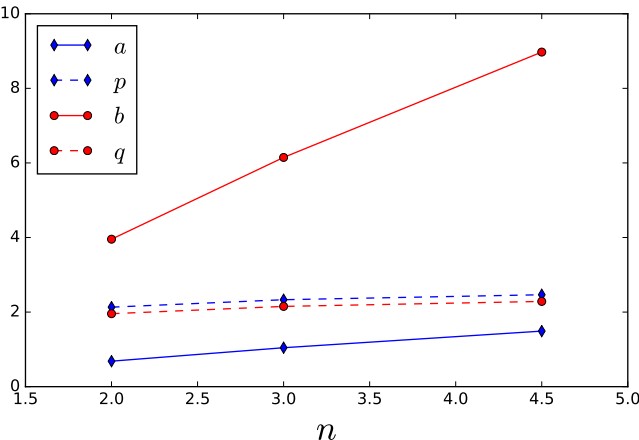

**Figure 7.** Evolution of the fitted coefficients $a$, $p$, $b$ and $q$ with respect to the exponent $n$ of the ice Norton Hoff constitutive behavior.

Another interesting feature which can be highlighted in Figure 4 is the fact that the isodissipation curves are closed for all the snow samples under consideration. This contrasts with the shape of ice isodissipation curves in the plane $(S_1/3, \bar{S}_2)$ which are represented by horizontal straight lines. Indeed, the corresponding viscous power doesn't depend on $S_1$ since the constitutive equation (2) for ice only involves the deviatoric stress $\overline{\sigma}$. The ability of snow to dissipate significant energy even under an isotropic loading ($\bar{S}_2 = 0$) is due to its porosity. Indeed, even under this type of macroscopic loading conditions, some regions of the microstructure experience a non zero deviatoric loading, which activates locally the ice viscous behavior. Even if this deviatoric loading vanishes on average, the mean viscous dissipated power only piles up, which results in closed isodissipation curves. Their shapes provide information about the ability of an isotropic macroscopic loading to locally activate the ice viscous behavior. Based on the Abouaf formulation (25), the ratio between the maximum isotropic stress $S_1^{\mathrm{max}}/3$ and the maximum deviatoric stress $\bar{S}_2^{\,\mathrm{max}}$ can be expressed for each sample as

$$\frac{(S_1^{\mathrm{max}}/3)}{\bar{S}_2^{\,\mathrm{max}}} = \sqrt{\frac{c}{6f}}. \tag{30}$$

As snow is always submitted to a mechanical loading which can be decomposed into a deviatoric part and an isotropic part, this ratio provides a measure of the relative contribution of the isotropic part of the mechanical loading in the activation of the ice viscosity. The bigger this ratio, the smaller the activation degree. The evolution of this ratio is plotted in Figure 8 as a function of snow compacity for $n \in \{2, 3, 4.5\}$. The diamond points are computed using the values for $f$ and $c$ presented in Table 3 and the solid line is computed using the two functions (29) with the parameters presented in Table 4. The increase in this ratio with snow density highlights the fact that deviatoric fluctuations get smaller under isotropic loading conditions as snow gets denser. In other words, the ice viscosity is more difficult to activate for dense snow than for fresh snow under isotropic loading conditions. The divergence of the solid line around 1 corresponds to the limit case of ice where $S_1^{\mathrm{max}}$ becomes infinite as predicted by (3). On this Figure, the dependence on the $n$ value is very limited, which highlights the fact that $n$ doesn't have

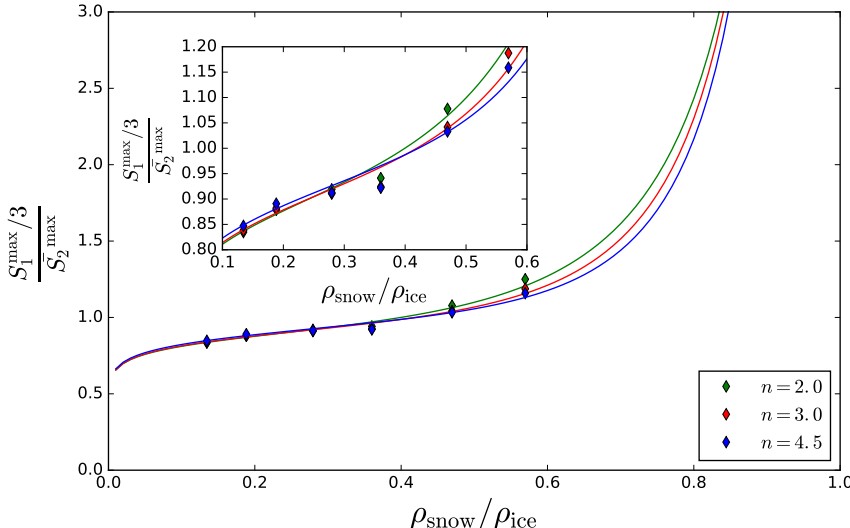

**Figure 8.** Aspect ratios $(S_1^{\max}/3)/S_2^{\max}$ of the Abouaf models with respect to snow compacity. Numerical results are represented with diamond points, results computed using functions (29) with solid lines.

any influence on the shape of the isodissipation curves but only on the stress level leading to a given isodissipation as already seen in Figure 5.

### 4.3 Normality rule

The proposed macroscopic modeling is formulated within the framework of associated viscoplasticity. In other words, the flow
5   direction $\dot{\mathbf{E}}$ is by construction supposed to be orthogonal to the isodissipation curves (see equation (13)). In the space composed of the two first stress and strain invariant planes, for an isodissipation curve corresponding to $\Sigma_{\mathrm{eq}}$, this normality is written

$$\begin{pmatrix} E_1 \\ \bar{E}_2 \end{pmatrix} \propto \begin{pmatrix} \dfrac{3\partial\Sigma_{\mathrm{eq}}}{\partial S_1} \\ \dfrac{\partial\Sigma_{\mathrm{eq}}}{\partial \bar{S}_2} \end{pmatrix}. \tag{31}$$

In the case of Abouaf models (see equation (25)), theoretical strain flow vectors associated to the seven points in Figure 4 are given as

10   $$\begin{pmatrix} E_1 \\ \bar{E}_2 \end{pmatrix} = A\,\Sigma_{\mathrm{eq}}^{n-1} \begin{pmatrix} 3f(\phi)\,S_1 \\ \tfrac{3}{2}c(\phi)\,\bar{S}_2 \end{pmatrix}. \tag{32}$$

In Figures 4 and 5, theoretical strain flow vectors are represented by dashed arrows for the radial projections of the numerical points on Abouaf fits. The overall comparison with their numerical counterparts represented by solid arrows is quite satisfactory, especially for the densest snows. However, concerning the flow direction, the Abouaf's model tends to over-predict the

strain deviatoric component for high deviatoric stresses $\bar{S}_2$ and to under-predict the strain deviatoric component for low deviatoric stresses except for the direction $\theta = 0°$. In terms of magnitude, the Abouaf's model tends to under-predict the intensity of the flow for deviatoric loading.

The observed difference between theoretical and numerical flow vectors actually results from the slight misfit between the Abouaf models and the numerical points, which is amplified by the radial projection procedure used in order to compute the theoretical flow vectors. Moreover, the validity of normality rule tends to get less accurate as the porosity of the material increases. A similar trend has been already observed in the case of power law fluid flow through porous media (Orgéas et al., 2007). Overall, the Abouaf's model presented in section 3.3 provides a satisfactory modeling of snow viscous behavior on the whole range of investigated densities.

## 5  Application to classical laboratory tests

In the case of isotropic snow microstructures, the homogenized constitutive viscous behavior developed in this paper can be summarized as

$$\dot{\mathbf{E}}_v = \dot{\mathrm{E}}_{\mathrm{eq}} \frac{\partial \Sigma_{\mathrm{eq}}}{\partial \boldsymbol{\Sigma}} = A \Sigma_{\mathrm{eq}}^{n-1} \left( f(\phi) S_1 \mathbf{I} + \frac{3}{2} c(\phi) \overline{\boldsymbol{\Sigma}} \right) \tag{33}$$

with

$$\Sigma_{\mathrm{eq}}(S_1, \bar{S}_2, \phi) = \sqrt{f(\phi) S_1^2 + \frac{3}{2} c(\phi) \bar{S}_2^2}, \qquad \dot{\mathrm{E}}_{\mathrm{eq}}(E_1, \bar{E}_2, \phi) = \sqrt{\frac{E_1^2}{9 f(\phi)} + \frac{2}{3} \frac{\bar{E}_2^2}{c(\phi)}}, \tag{34}$$

and

$$f(\phi) = a \left( \frac{\phi}{1-\phi} \right)^p, \qquad c(\phi) = 1 + b \left( \frac{\phi}{1-\phi} \right)^q \tag{35}$$

where $n$ and $A$ account for the ice viscosity (Table 1) and $a$, $p$, $b$ and $q$ account for snow porosity (Table 4).

In the following, the mechanical responses of the proposed model are analyzed and compared in the case of classical laboratory tests. In Figure 9, situation (a) corresponds to an oedometric compression test in which the radial deformation $E_{rr}$ of the snow sample is prevented. Snow mechanical response is then characterized by the relationship between the axial stress $\Sigma_{zz}$ and the axial strain rate $\dot{E}_{zz}$. Situation (b) corresponds to a general triaxial compression test in which the radial stress $\Sigma_{rr}$ is prescribed and kept constant. From this general setting, two particular cases can be studied: a uniaxial compression test when $\Sigma_{rr} = 0$ and an isotropic compression test when $\Sigma_{zz} = \Sigma_{rr}$. In all this section, classical soil mechanics convention is adopted, i.e. compression stresses are positive, and snow elasticity is neglected.

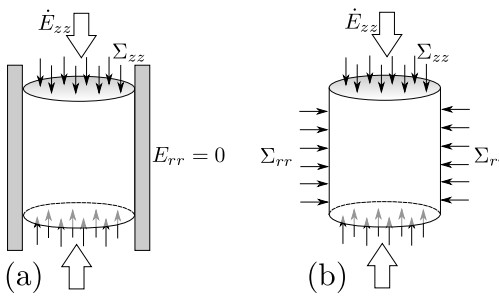

**Figure 9.** Oedometric compression test (a) and triaxial compression test (b).

## 5.1 Oedometric compression test

As the snow samples are often extracted from the snowpack thanks to hollow cylinders, the oedometric compression test is one of the most convenient mechanical laboratory test to perform on snow. Under the lateral constraint $E_{rr} = 0$, we have

$$
\dot{\boldsymbol{E}} = \begin{pmatrix} 0 & 0 & 0 \\ 0 & 0 & 0 \\ 0 & 0 & \dot{E}_{zz} \end{pmatrix}.
\tag{36}
$$

5    The quasi-static equilibrium $\mathrm{div}\boldsymbol{\Sigma} = 0$ implies that $\Sigma_{\theta\theta} = \Sigma_{rr}$. Consequently, the macroscopic stress tensor is written

$$
\boldsymbol{\Sigma} = \begin{pmatrix} \Sigma_{rr} & 0 & 0 \\ 0 & \Sigma_{rr} & 0 \\ 0 & 0 & \Sigma_{zz} \end{pmatrix}.
\tag{37}
$$

As a result, the two first strain rates and stress invariants are written

$$
E_1 = \dot{E}_{zz}, \qquad \bar{E}_2 = \sqrt{\frac{2}{3}}\dot{E}_{zz}, \qquad S_1 = 2\Sigma_{rr} + \Sigma_{zz}, \qquad \bar{S}_2 = \sqrt{\frac{2}{3}}(\Sigma_{zz} - \Sigma_{rr}), \quad \text{with } \Sigma_{zz} \geq \Sigma_{rr}.
\tag{38}
$$

In this particular case, from (33) and (34), it can be shown that the lateral constraint $E_{rr} = 0$ implies that

$$
\frac{\Sigma_{rr}}{\Sigma_{zz}} = \frac{c(\phi) - 2f(\phi)}{c(\phi) + 4f(\phi)},
\tag{39}
$$

and consequently,

$$
\Sigma_{\mathrm{eq}} = \sqrt{\frac{9c(\phi)f(\phi)}{4f(\phi) + c(\phi)}}\Sigma_{zz}, \quad \text{and} \quad \dot{E}_{zz} = A\left(\frac{9f(\phi)c(\phi)}{4f(\phi) + c(\phi)}\right)^{\frac{n+1}{2}}\Sigma_{zz}^n.
\tag{40}
$$

In oedometric experimental tests, the lateral pressure $\Sigma_{rr}$ is not easily accessible and it is often tempting to neglect this pressure and interpret any oedometric compression test as a uniaxial compression test. The relation (39) can be used to assess the relative

15   importance of the confining pressure with respect to the vertical stress. The evolution of this ratio with respect to snow density

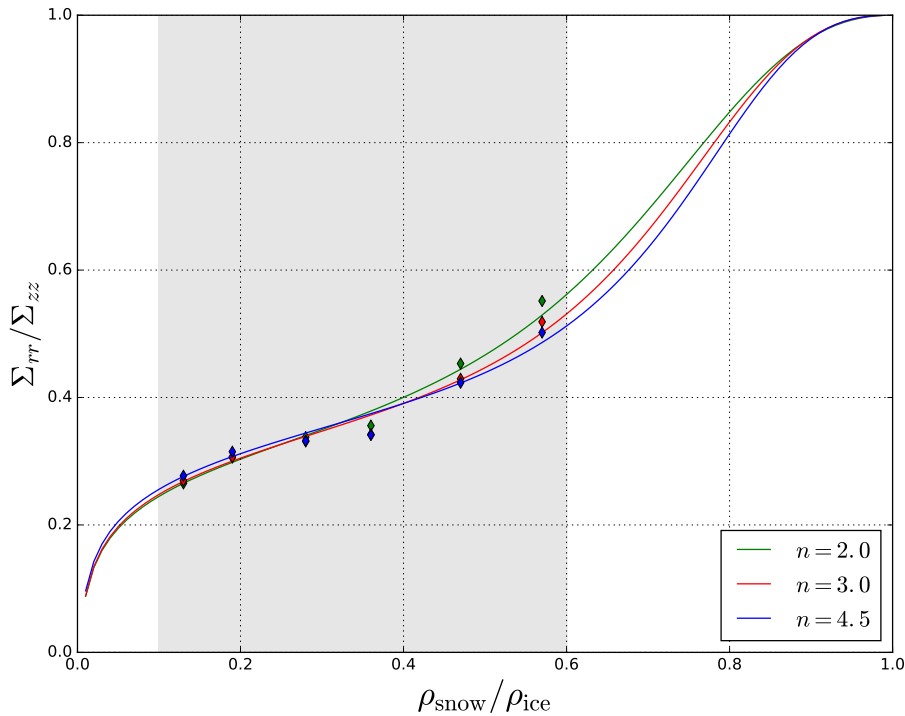

**Figure 10.** Evolution of the ratio between the lateral and axial stresses with respect to snow compacity during any oedometric compression test for different values of the exponent $n$ (numerical points as diamond points, results computed using functions (29) as solid lines). The compacity range for typical snow samples is materialized by the gray zone.

is shown in Figure 10 for $n \in \{2, 3, 4.5\}$. It should be noticed that this ratio does not depend on the axial strain rate $\dot{E}_{zz}$. As expected, this ratio increases with increasing snow density, and tends towards one for $\phi = 0$, due to the incompressiblity of the ice skeleton. In the whole range of snow compacities under consideration (materialized by the gray zone in Figure 10) the lateral pressure represents 30 to 50 % of the vertical stress and cannot be neglected in practice. Finally, let us remark that the evolution of this ratio is similar to the one measured by Geindreau et al. (1999a) and Viot and Stutz (2002) on metallic powders. Figure 10 also shows that this ratio is almost independent of the exponent $n$, which is consistent with the experimental results of Viot and Stutz (2002).

## 5.2 Triaxial compression test

During a triaxial test, a cylindric snow sample is submitted simultaneaously to an axial stress $\Sigma_{zz}$ and a lateral confining pressure $\Sigma_{rr}$. The static equilbrium $\mathrm{div}\,\mathbf{\Sigma} = 0$ implies that $\Sigma_{\theta\theta} = \Sigma_{rr}$, consequently the macroscopic stress tensor is written

$$\mathbf{\Sigma} = \begin{pmatrix} \Sigma_{rr} & 0 & 0 \\ 0 & \Sigma_{rr} & 0 \\ 0 & 0 & \Sigma_{zz} \end{pmatrix}. \tag{41}$$

In these conditions, the constitutive equation (33) implies that $\dot{E}_{\theta\theta} = \dot{E}_{rr}$, and thus

$$\dot{\mathbf{E}} = \begin{pmatrix} \dot{E}_{rr} & 0 & 0 \\ 0 & \dot{E}_{rr} & 0 \\ 0 & 0 & \dot{E}_{zz} \end{pmatrix}. \tag{42}$$

As a result, the two first strain rate and stress invariants are written

$$E_1 = 2\dot{E}_{rr} + \dot{E}_{zz}, \qquad \bar{E}_2 = \sqrt{\frac{2}{3}}(\dot{E}_{zz} - \dot{E}_{rr}), \qquad S_1 = 2\Sigma_{rr} + \Sigma_{zz}, \qquad \bar{S}_2 = \sqrt{\frac{2}{3}}|\Sigma_{zz} - \Sigma_{rr}|. \tag{43}$$

In this particular case, from (33) and (34) it can be shown that:

$$\Sigma_{\mathrm{eq}} = \sqrt{f(\phi)(2\Sigma_{rr} + \Sigma_{zz})^2 + c(\phi)(\Sigma_{zz} - \Sigma_{rr})^2}. \tag{44}$$

and

$$\begin{cases} \dot{E}_{rr} = A\Sigma_{\mathrm{eq}}^{n-1}\left[\left(2f(\phi) + \frac{1}{2}c(\phi)\right)\Sigma_{rr} + \left(f(\phi) - \frac{1}{2}c(\phi)\right)\Sigma_{zz}\right] \\ \dot{E}_{zz} = A\Sigma_{\mathrm{eq}}^{n-1}\left[(2f(\phi) - c(\phi))\Sigma_{rr} + (f(\phi) + c(\phi))\Sigma_{zz}\right] \end{cases}. \tag{45}$$

In the case of a uniaxial compression test, $\Sigma_{rr}$ must be set to 0 in the above equations.

In order to compare the mechanical response of snow under various loadings (uniaxial, oedometric, isotropic and triaxial tests), Figure 11 presents the evolution of the snow densification rate (for $n = 4.5$ and $A = 1.5\ 10^{-3}\,\mathrm{MPa^{-n}.s^{-1}}$) given by $E_1 = \dot{\rho}_{\mathrm{snow}}/\rho_{\mathrm{snow}}$ with respect to snow compacity when constant stresses are applied on the sample. As expected, this figure shows that:

- whatever the loading, the densification rate strongly decreases with increasing snow density. In the investigated range, i.e. $\rho_{\mathrm{snow}}/\rho_{\mathrm{ice}} \in [0.1, 0.6]$, the densification rate decreases by 9 orders of magnitude from $10^{-1}\mathrm{s^{-1}}$ to $10^{-10}\mathrm{s^{-1}}$.

- for a given snow density, the loading conditions influence strongly the densification rate. Typically, when $\Sigma_{zz} = 10$ kPa the densification rate decreases by nearly one order of magnitude if the confining pressure $\Sigma_{rr}$ is reduced from 10 kPa (isotropic compression) to 0 kPa (uniaxial compression). On the contrary, when $\Sigma_{rr} = 10$ kPa, the densification rate increases by nearly one order of magnitude if the axial stress $\Sigma_{zz}$ increases from 10 kPa (isotropic compression)

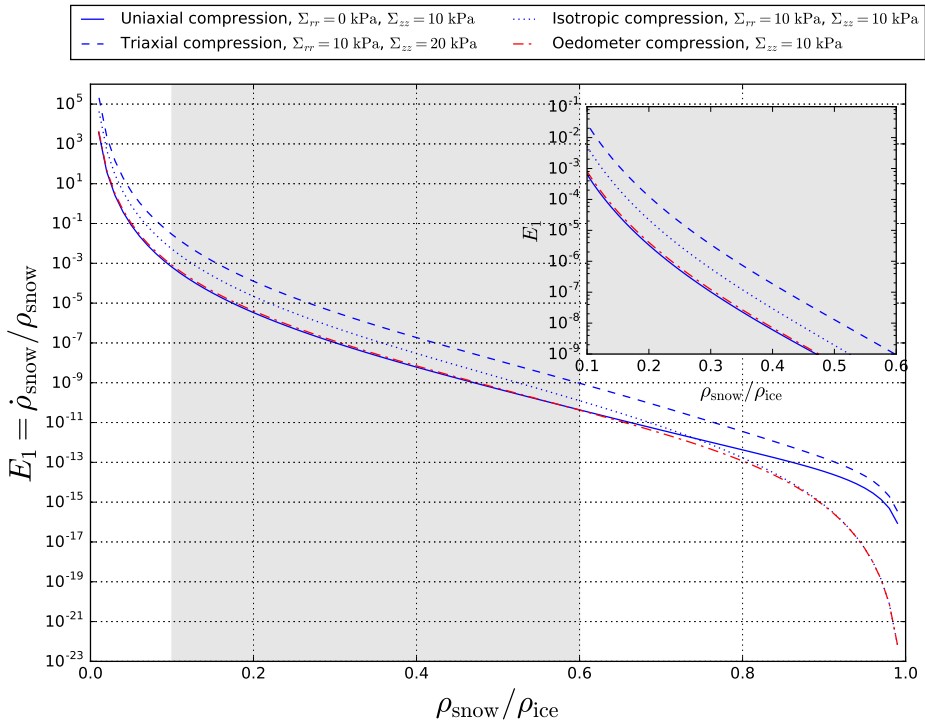

**Figure 11.** Predicted densification rate (for $n = 4.5$ and $A = 1.5\ 10^{-3}\,\mathrm{MPa^{-n}.s^{-1}}$) with respect to snow compacity for four classical laboratory tests: an oedometric compression test (dash-dotted line), a uniaxial compression test (solid line), a triaxial compression test (dashed line) and an isotropic compression test (dotted line). The inset plot provides a zoom on the classical range of snow densities observed experimentally (gray background).

to $20$ kPa (triaxial compression). As expected, this last result shows the increase in the densification rate with the increase in the deviatoric stress (i.e. $\bar{S}_2$). Even if the lateral confining pressure cannot be neglected during oedometric test as highlighted in Figure 10, the oedometric compression test results in a similar densification rate as the uniaxial compression test for the same axial stress $\Sigma_{zz}$. Indeed, the vertical strain rate is lower for an oedometric compression than for a uniaxial one but the geometrical constraint imposed in the oedometric compression test prevents the snow
5      sample from dilating, which is not the case for the uniaxial compression test. Overall the two effects cancel out each other. Finally, above the classical snow compacity range ($\rho_{\mathrm{snow}}/\rho_{\mathrm{ice}} \geq 0.6$), the densification rate dramatically decreases for the oedometric and isotropic compression tests due to the ice incompressibility. As already underlined in Figure 10 for this limit case, the oedometric compression test is equivalent to the isotropic compression one.

10      In practice, the strain rate is often imposed on the sample. The Figure 12 presents the evolution of the stress $\Sigma_{zz}$ versus snow compacity $\rho_{\mathrm{snow}}/\rho_{\mathrm{ice}} = 1 - \phi$ for two different values of strain rates $\dot{E}_{zz} \in \{10^{-7}; 10^{-5}\}\ \mathrm{s^{-1}}$ and the different loading conditions (uniaxial, oedometric, isotropic and triaxial tests). This figure suggests the following comments:

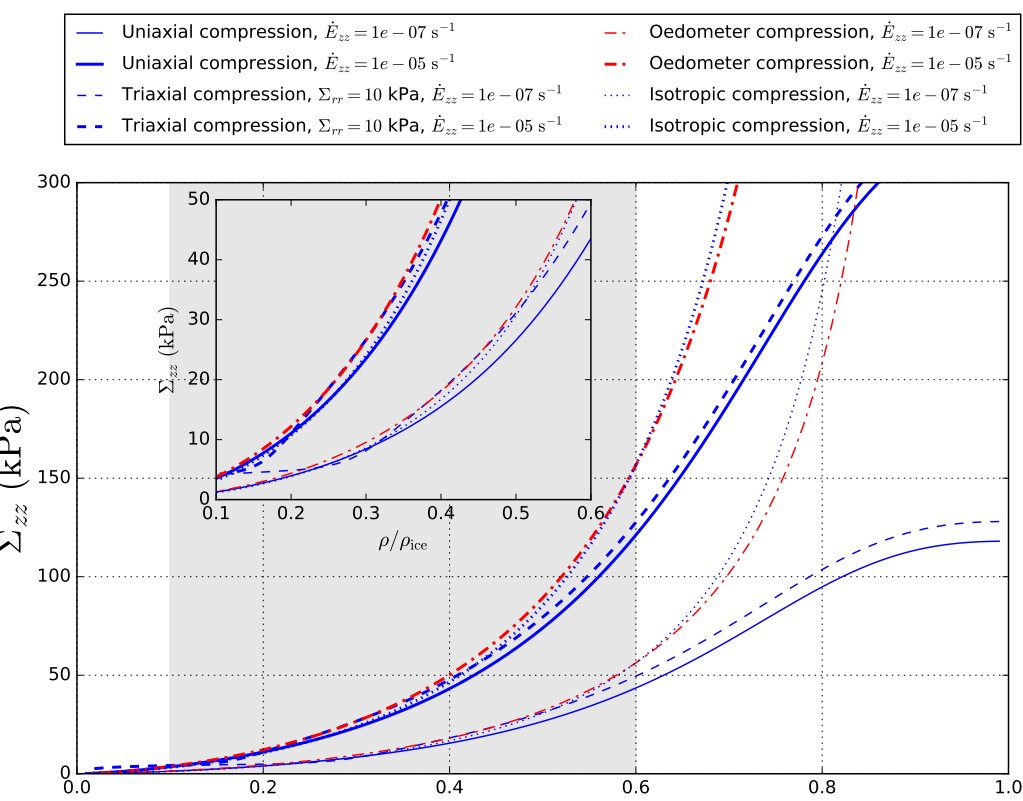

**Figure 12.** Predicted axial stress level (for $n = 4.5$ and $A = 1.5 \; 10^{-3} \, \mathrm{MPa^{-n}.s^{-1}}$) with respect to snow compacity for two imposed strain rates (line thickness) and for four classical laboratory tests: an oedometric compression test (dash-dotted line), a uniaxial compression test (solid line), a triaxial compression test (dashed line) and an isotropic compression test (dotted line). The inset plot provides a zoom on the classical range of snow densities observed experimentally (gray background).

- as expected, for a given strain rate, the stress $\Sigma_{zz}$ increases with increasing snow density.

- for a given strain rate and a given density, the stress $\Sigma_{zz}$ increases with increasing the lateral pressure $\Sigma_{rr}$ around the sample.

- for a given snow density, the stress $\Sigma_{zz}$ strongly increases with increasing strain rate, which is in accordance with the power law relationship.

- the ice viscoplastic behavior is recovered when $\rho_{\mathrm{snow}}/\rho_{\mathrm{ice}}$ tends towards 1. For a given strain rate, the axial stress for a uniaxial or triaxial compression test tends towards a maximum value. By contrast, due to ice incompressibility, the axial oedometer stress $\Sigma_{zz}$ tends towards $+\infty$.

In order to quantitatively compare the predictions of our model against the experimental results of Bartelt and von Moos (2000), we consider a snow of density $\rho_{\text{snow}} = 255$ kg.m$^{-3}$ (corresponding to $\rho_{\text{snow}}/\rho_{\text{ice}} = 0.27$) subjected to a confining pressure $\Sigma_{rr} = 2.5$ kPa and a strain rate $\dot{E}_{zz} = 2.2 \times 10^{-5}$ s$^{-1}$. In this case, the axial stress predicted by our model is $\Sigma_{zz} = 22.8$ kPa, which is consistent with the experimental values obtained by Bartelt and von Moos (2000) around 30 kPa.

Further comparison with Bartelt and von Moos (2000) can be achieved in the case of a uniaxial compression test ($\Sigma_{rr} = 0$). In this case, the axial stress simply reads

$$\Sigma_{zz}(\dot{E}_{zz}, \phi) \;=\; \left( \frac{\dot{E}_{zz}}{A(f(\phi) + c(\phi))^{\frac{n+1}{2}}} \right)^{\frac{1}{n}}. \tag{46}$$

For a given strain rate, the mechanical response of snow can be compared to the one of ice as in Bartelt and von Moos (2000) by using the following parameter:

$$\alpha_\eta(\phi) \;=\; \frac{\Sigma_{zz}^{\text{snow}}}{(1-\phi)\Sigma_{zz}^{\text{ice}}} \;=\; \frac{\Sigma_{zz}(\dot{E}_{zz}, \phi)}{(1-\phi)\Sigma_{zz}(\dot{E}_{zz}, 0)} \;=\; \frac{1}{1-\phi} \left( \frac{1}{f(\phi) + c(\phi)} \right)^{\frac{n+1}{2n}}. \tag{47}$$

This parameter compares the axial stress that a given snow sample can transmit ($\Sigma_{zz}^{\text{snow}}$) to a rough estimate of this stress given as a fraction of the axial stress transmitted in the case of ice ($(1-\phi)\Sigma_{zz}^{\text{ice}}$). In Figure 13, the above theoretical expression of $\alpha_\eta(\phi)$ is compared with the experimental fit $\alpha_\eta = 0.0028 \exp(0.008 \; \rho_{\text{ice}} \; (1-\phi))$ proposed by Bartelt and von Moos (2000). As expected, $\alpha_\eta(\phi)$ increases with increasing snow density. By definition, $\alpha_\eta(\phi)$ should vary between 0 and 1. We
can observe that the theoretical expression of $\alpha_\eta(\phi)$ is strictly greater than 1 for $\rho_{\text{snow}}/\rho_{\text{ice}} \in [0.8, 1]$, which is not physically reasonable. This feature results from the independent choices of the parameters $a$, $b$, $p$ and $q$ in the fitting procedure used in subsection 4.2. An implicit relation between these parameters could help in order to ensure that $\alpha_\eta(\phi)$ remains lower than 1 in the whole compacity range. Nevertheless, in the range of snow densities under consideration ($\rho_{\text{snow}}/\rho_{\text{ice}} \in [0.1, 0.6]$), $\alpha_\eta$ increases monotonously between roughly 0.1 and 0.6. This prediction is higher than the experimental fit proposed by Bartelt
and von Moos (2000) (see Figure 13). However, during this experiment, the macroscopic mechanical response probably results from both viscous deformation of ice skeleton and ruptures of ice bridges between snow grains. Even if our model is able to account for some microstructure modifications through the porosity dependence of the parameters $f$ and $c$, the changes induced by the experimental testing conditions might exceed the scope of application of our model.

## 6   Conclusions

Despite the non-linearity of the ice viscous constitutive equation, the image-based homogenization approach introduced by Wautier et al. (2015) was successfully adapted to the numerical homogenization of snow viscous behavior. It allows the viscoplastic response of any snow sample being computed from its X-ray tomographic image. By contrast to the elastic case, the macroscopic stress response is not a linear function of the imposed macroscopic strain anymore. As a result, snow macroscopic response was investigated in terms of isodissipation curves in the planes of the two first strain rate and stress invariants. The
shape and size of these curves characterize the strong coupling between the snow microstructure and the ice viscous behavior at

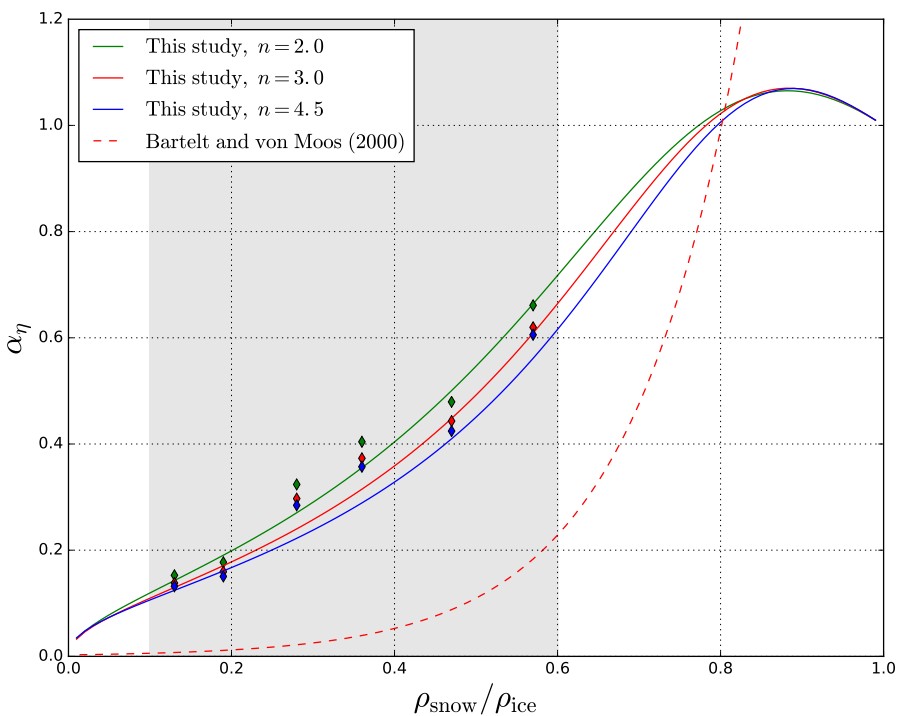

**Figure 13.** Comparison between our theoretical prediction for $\alpha_\eta$ (47) and the experimental fit proposed by Bartelt and von Moos (2000). The points computed directly from the values for $f$ and $c$ reported in Table 3 are shown as diamond points. The compacity range for snow samples under consideration is materialized by the gray zone.

the microscopic scale (power law with an exponent $n$). Different values for $n$ were considered in this study to give some insight on their influence on the homogenized viscoplastic behavior of snow. Thanks to a few selected loading paths, an Abouaf model was fitted onto numerical simulation results. This formulation seems to be relevant to describe the snow viscoplastic behavior in the whole range of snow density under consideration ($0.43 < \phi < 0.87$), provided that snow microstructure is isotropic. For 5    a given value of $n$, the influence of the snow microstructure on its viscoplastic response is described at the first order through two material functions $f(\phi)$ and $c(\phi)$ (see (29)) depending on the porosity only.

The robustness of this Abouaf formulation was tested for several isotropic snow samples covering the whole range of accessible densities. The fitted models proved to be able to account for the stress and strain rate levels as well as the viscous flow directions. In particular and contrary to the case of ice, the ability of snow to exhibit a viscous behavior even under isotropic strain loading 10    is recovered. The scope of application of the presented unified formulation is quite promising and could help improve the modeling of the densification of the snowpack in avalanche forecasting models.

The proposed homogenization model can be easily used to predict the viscous behavior of snow in classical laboratory tests as illustrated in the last section of this paper. However, the uncertainties made on our model parameters should be quantified

through a sensitivity analysis, in order to reckon the ability of our homogenized law for snow viscosity to quantitatively recover the experimental results of Desrues et al. (1980); Bartelt and von Moos (2000); Moos et al. (2003); Scopozza and Bartelt (2003b).

Even though the porosity is known to have a very strong influence on the resulting homogenized properties of snow, it is

also acknowledged that the very strong anisotropy of some snow microstructures cannot be neglected. The importance of this anisotropy was quantified within the framework of elasticity (Srivastava et al., 2010, 2016; Wautier et al., 2015) but the extension of our homogenized visco-plastic formulation to anisotropic snow types is quite challenging as the dimension of the invariant space will increase dramatically (Boehler, 1978; Liu, 1982; Hansen et al., 1991).

Finally, in the present work based on FEM simulations, the ice skeleton is viewed as a continuous polycrystalline material.

The proposed methodology to identify and formulate the 3D viscoplastic behavior of snow could be also applied to DEM simulations. For that purpose, the identification of the shape and crystalline orientations of every ice grain (Rolland du Roscoat et al., 2011) as well as the knowledge of the viscoplastic contact laws (Burr et al., 2015b, a) would be of primary interest.

*Acknowledgements.* We thank S. Lejeunes and S. Bourgeois from the LMA for sharing with us the code of their plugin Homtools and for helping us to use it in our scripts. Tomographic images have been acquired at ESRF (ID19) and 3SR Lab. CNRM/CEN is part of the

LabEx OSUG@2020 (ANR10 LABX56). 3SR lab is part of the LabEx Tec 21 (Investissements d'Avenir - grant agreement ANR11 269 LABX0030). The authors gratefully acknowledge the anonymous reviewers, Andrew Hansen and Maurine Montagnat for their comments and advices, which helped improve the quality of the final manuscript.

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
