# Peer review of "Numerical homogenization of the viscoplastic behavior of snow based on X-ray tomography images"

_The Cryosphere, 2016_

## Referee Comment (RC1) · Anonymous Referee #1 · 20 Dec 2016

Reviewer comments on: Numerical homogenization of the viscoplastic behavior of snow based on X-ray tomography images

Authors Antoine Wautier, Christian Geindreau, and Frédéric Flin

Background The authors apply a technique whereby information on the 3D microscopic structure of snow obtained from X-ray tomography can be "homogenized" to develop a macroscale theory of snow viscoplastic deformation under various loading regimes. Homogenization involves taking the inhomogeneous 3D microstructure of a volume of snow that is sufficiently large such that the variations in mechanical behavior at the microscopic scale are averaged over the volume element to yield mechanical behavior that is representative of the whole (representative volume element; RVE), or contin-

uum scale, material. It is a method to incorporate the influences of microscopic scale structure, properties and processes into a continuum description of the material. For snow, this technique was applied by Wautier et al. (2015) to estimate the elastic RVE properties using the homogenization process (i.e., converting the 3D image of snow microstructure into a gridded finite element model and applying loading at the outer boundaries of the RVE sample and determining the averaged elastic deformation response over the sample volume. This works well for elastic response, as the elastic compliance matrix is general enough to account for anisotropic behaviors, which are identified in the homogenization process. For non-elastic behavior the situation is much more challenging as the initial microscopic structure is only the starting point of deformation and it is necessary to develop an deformation evolution algorithm as a function of loading/strain rate for the RVE to use in the continuum scale model.

The authors use the homogenization process and laboratory tests on ice and snow to develop parameters for their adaptation of a viscoplastic model (Abouaf, 1985) then compare the model output to their laboratory test results with generally good agreement. They also make further qualitative comparison to data from Bartelt and von Moose (2000).

Review Efforts to develop viscous models for snow deformation have been ongoing since the earliest days of snow mechanical properties studies (the authors list but a few). So far the models all have two things in common, they are able to well represent laboratory test data yet are unable to describe actual snow behavior in a general way, even with the increasing model complexity that has developed over time. The reason for this is that most viscous, viscoelastic, and viscoplastic models consist of complex functions with numerous free parameters that can be adjusted to obtain theoretical agreement for even the most complex observed behavior. The difficulty is that the models lack accurate physics and are essentially empirical or semi-empirical by design. This works well for the purpose that the models are generally developed, which is to describe engineered materials with well-defined and repeatable material behavior.

[Figure]

This does not mean that such models are not useful for representing snow behavior in limited situations. When one has some confidence that model parameterizations will produce behavior within known limits and acceptable engineering accuracy over a limited time period the usefulness of viscous, viscoelastic and viscoplastic models as an engineering tool has been demonstrated. To develop continuum theories that can provide more accurate predictive behavior of snow deformation over longer time periods that capture the influence of microscopic structure, mechanical properties, and deformation processes it is necessary to get the microscale physics right during the homogenization process. While the authors recognize the importance in improving the microscale physics they do not do so in this work. The difficulty in relating microscale structure and processes to continuum scale is one that the research community studying granular materials is also struggling with (Walsh et al., 2007).

There are several fundamental difficulties with the current paper: 1. The authors indicate that the macroscopic mechanical behavior of snow strongly depends on its microstructure, which is mainly characterized by its density, topology, external loading (elastic, visco-plastic, brittle-failure) temperature and applied strain-rate. What is missing from the list are the mechanical processes that occur at the microscale (e.g., grain boundary sliding, bond breakage, bond sintering and re-sintering after breakage and particle rearrangement; ignoring metamorphic changes) that are critical to define the evolution of the snow under defined loading (or strain rate) conditions in the RVE. Knowledge of the microstructure is only the starting point to define an evolution function. It is the explicit microscale mechanical processes that determine how snow deformation evolves within the RVE and the model described in the current work does not, and cannot, accurately represent the microscale mechanical processes necessary to develop and RVE evolution function. 2. The Abouaf (1985) viscoplastic model used in the current work appears to be primarily porosity dependent and the authors do not provide a clear description of how the homogenized snow microstructure is incorporated into the model. The free parameters of the model include four parameters in two porosity dependent functions, neither of which appear to incorporate microscale mechanical

processes. There are continuum scale models specifically developed to incorporate observed micro-scale measurement that might prove more suitable to this effort, if an appropriate evolution function can be determined. 3. The finite element method applied in the paper is not capable of accurately simulating the important micro-scale mechanical processes involved in the RVE deformation (primarily, grain boundary sliding and particle rearrangement). The only techniques that have demonstrated the ability to represent the micro-scale processes in snow to-date are the discrete element method (DEM) (Johnson and Hopkins, 2005; Hagenmuller et al., 2015) and the deformable DEM (Johnson, 1998). The authors mistakenly indicate that DEM models are only useful for simulating grain breakage and grain rearrangement. With appropriate interaction rules at the grain level the DEM is capable of modeling pretty much any physical process limited only by computational capability, which continue to improve at a rapid rate.

Reference Abouaf, M. (1985). Modélisation de la compaction de poudres métalliques frittées, Ph.D. thesis, 1985.

Bartelt, P. and von Moos, M. (2000). Triaxial tests to determine a microstructure-based snow viscosity law, Annals of Glaciology, 31, 457–462.

Hagenmuller, P., Chambon, G., and Naaim, M. (2015). Microstructure-based modeling of snow mechanics: a discrete element approach, Cryosphere, 20 9, 1969–1982,

Johnson, J. B. and M. A. Hopkins (2005). "Identifying microstructural deformation mechanisms in snow using discrete element modeling." Journal of Glaciology 51: 432-442.

Johnson, J. B. (1998). "A preliminary numerical investigation of the micromechanics of snow compaction." Annals of Glaciology 26: 51-54.

Walsh, S. D. C., et al. (2007). "Development of micromechanical models for granular media." Granular Matter 9: 337-352.

Wautier, A., Geindreau, C., and Flin, F. (2015). Linking snow microstructure to its macroscopic elastic stiffness tensor: A numerical homogenization method and its application to 3-D images from X-ray tomography, Geophysical Research Letters, 42, 8031–8041.
* * *

---

## Referee Comment (RC2) · Anonymous Referee #2 · 7 Jan 2017

I must confess that I am not sufficiently expert in theoretical continuum mechanics to be a useful referee; however, I will still accept my role as a referee for this paper. I trust the editor will understand my shortcomings and be able to use my comments in a constructive manner that will ultimately be useful for the eventual publication of the science involved.

After reading the abstract, introduction and conclusion, looking at the various figures, and skimming the somewhat "over my head" mathematical development, I am still quite confused about what this paper is about and what it is trying to do. I take it that there is an ultimate goal of providing a simple rule (or demonstration of a rule?) known as "homogenization" that, when applied to snow, allows the treatment of the tortuously

complex micro-structure of the snow grain assemblage to be done as if the whole mess were a single homogeneous fluid. I struggled to see where the paper demonstrates either success or failure (and by what criteria?) in achieving this overall goal.

I also found the paper to be very loosely tied to a snow phenomena that I am familiar with. Of course, not being a specific snow scientist, but rather a generalist in glaciology, I could be simply too unaware of what is specific to the field. But usually, I get hints from reading the gist of the introduction and abstract as to what the specific phenomena to be illuminated is. I did not get such a hint in this paper.

Finally, I don't understand why viscoelasticity is being examined. Aren't the two end member rheological treatments (pure viscous for long-term creeping problems, or pure elasticity for short term shock events) good enough for practical snow problems? What is the motivation for developing a viscoelastic treatment? What problem needs it?

The demonstration is also somewhat opaque due to the fact that a black-box software tool (ABAQUS) is used for much of the computation... Some of this computation is non trivial...

The paper reads very much like a textbook; and I wonder how much of the mathematical development is simply boilerplate that is also published elsewhere.

Good luck with this paper! It is obviously written by incredibly smart scientists, and there is a lot of work there. I simply don't know if I have learned anything from reading it.
* * *

---

## Referee Comment (RC3) · A.C. Hansen (Referee) · 10 Jan 2017

Review of:

**Numerical homogenization of the viscoplastic behavior of snow**

**based on X-ray tomography images**

January 2017

The authors of this paper develop a nonlinear viscoplastic constitutive law for snow that is the result of viscoplastic behavior of the ice constituent. The process involves what I'll refer to as a finite element micromechanics model of a representative elementary volume (REV) whose microstructure is determined from snow samples using X-ray tomography. The micro stress-strain fields are then appropriately homogenized to aid in the development of a macroscale viscoplastic stress-strain relation. The resulting macroscale constitutive law appears relative straightforward to implement and contains a modest number of empirical parameters—as does any nonlinear constitutive model of interest.

I believe the authors have addressed a very challenging problem and produced a constitutive model for snow that is of interest. The paper is well written and I was able to follow the majority of the work. Admittedly, there were also areas of the article where I simply did not have the expertise to follow.

I believe the article is worthy of publication although I'll provide several comments and/or suggestions for revisions in the interest of improving the manuscript and perhaps expanding the readership. I'll also provide some minor editorial comments.

**Comments/Suggestions:**

1. I was misled by the title. When I first read it, I thought of a finite (large) strain viscoplastic constitutive model for snow where the important deformation mechanisms include such things as bond fracture, intergranular glide, neck growth, etc. That is clearly not the topic of the article. Perhaps the title could be modified to reflect this point by simply adding words such as "*small strains*" or some other appropriate descriptor. I'll note that this point is a recurring theme in my review.

2. Page 1 lines 15-18 Echoing the point above, lines 15-18 contain a sentence: "*In practice, a good knowledge of the macroscopic mechanical behavior of snow in a wide range of applied loads, strain rates, and temperatures is of particular interest with respect to avalanche risk forecasting or to determine the forces on avalanche defense structures.*"

   This sentence led to further confusion, for me, in that I again immediately thought of finite deformation (large strain) problems where constitutive modeling is an enormous and important challenge.

   In reality, this paper is concerned with very small strains, and perhaps small strain rates, where bond fracture and intergranular glide do not occur. More specifically, the original ice microstructure must be intact. I believe this constraint is pretty severe for low density snow where bond fracture will surely initiate at very low strains.

   I think the reader would be well served if the authors clarify this point by adding a paragraph in the introduction describing the range of applicable strains where they believe the theory is

valid. Quantifying such strain levels may be hard to do so an alternate approach would be to simply state that the microstructure is assumed unaltered by bond fracture and further describe the types of problems one might address with the theory. For example, density consolidation under the external body force of gravity seems to me to be the topic of most relevance.

3. Page 3 line 3 It would be helpful, at least to me, to provide a brief description of *isodissipation curves* and how they will be utilized. Are these curves analogous to a yield surface in rate-independent plasticity? This is an area of the manuscript where I was a bit lost and I suspect others will suffer a similar plight.

4. Page 3 line 30. It would be useful to point out here that **E** is the "small strain tensor" at the macroscale. Again, I was initially confused as in finite strain theory, **E** often refers to the Lagrangian finite strain measure. Moreover, finite strain problems are commonplace in snow mechanics.

5. Page 3 lines 16-27 The discussion of the various types of boundary conditions is excellent and I completely agree with the authors. I suspect a formal study of the differences between periodic boundary conditions and kinematically uniform boundary conditions would show very little difference in a homogenization process. The volume averaging process is very forgiving.

6. Page 7 line 3 The focus of this paper is on assumed isotropic behavior (I believe that is a good thing!). Indeed, on line 1 of page 8 this is explicitly stated. Hence, on page 7, line 3 it might be useful to state that, for an isotropic material, the stiffness tensor can be obtained with a single simulation of an REV. Six simulations are sufficient to fully characterize orthotropic behavior.

7. Page 7 lines 12-13 I was completely lost by the phrase "*unit sphere in the second order tensor space.*"

8. Page 9, lines 16-17 I don't believe this sentence is properly expressed. While there is certainly an elastic strain and a viscous strain, there is only one stress. That is, we do not have an elastic stress and a viscous stress. The equation for stress is correct in these lines, the wording is not.

9. Page 15 line 8 "The ability of snow to dissipate some energy …. " Perhaps it would be better to say: "The ability of snow to dissipate significant energy …. " A minor point perhaps but a reflection of my background.

10. Page 18 lines 11-14 I agree with the components identified in the strain tensor, all zero except $E_{zz}$. However, I am confused by the stress tensor. Specifically, the authors show $\Sigma_{\theta\theta} = \Sigma_{rr}$. While this may be correct, it is not readily apparent to me. Perhaps the authors could show this point, at least in their review response if not the manuscript.

11. Page 20 lines 1-4 Similar comments to the previous point. In the case of stress, it is unclear why $\Sigma_{\theta\theta} = \Sigma_{rr}$. For example in a thick walled pressure vessel $\Sigma_{\theta\theta}$ is vastly different than $\Sigma_{rr}$.

Moreover, why is $E_{\theta\theta} = E_{rr}$? My instincts tell me that this problem is over-constrained and that these two conditions involving stress and strain cannot be met simultaneously. Again, perhaps I am wrong but additional clarity is needed.

12. Page 23 lines 16-18 This comment is important and follows back to my earliest comments about limitations of the theory to very small strains. The authors use the phrase "*few percents of deformation*." I'm guessing even strains of this low magnitude may be too high to preclude bond breaking, particularly at low densities.

13. Page 24 lines 8-11 I respectfully disagree with the authors as to the importance of this study being extended to anisotropic snow types. In my view, there are so many more important avenues of discovery. I'll offer up a few suggestions of personal interest:

   a. Further validation of the theory with experimental data would certainly be a worthy endeavor. Once that is complete, applying the theory to a 2-D boundary value problem of natural consolidation (at the macroscale) would be of genuine interest. Even a field study comparison is possible.

   b. Of course, consolidation (densification) in snow will also occur due to sintering, particularly in equitemperature environments. It would be of interest to explore the significance of these two phenomena, particularly as a function of snow depth and temperature.

   c. Extending this work to finite deformation by allowing for microstructural evolution of key state variables such as grain size, intergranular glide paths, neck growth, etc. In this vein, I would point the authors to some of the pioneering work of R. L. Brown on viscoplastic behavior of snow. One particular reference of interest:

   Brown, R.L., "A volumetric constitutive law for snow based on a neck growth model" **Journal of Applied Physics**, Volume 51, 1980

   **Abstract**
   A volumetric constitutive law for snow is developed by considering the deformation of the ice grains and grain bonds which form the porous material. The equations of equilibrium and mass conservation are applied to both the grain body and neck regions to calculate the rate of change of grain geometry and neck geometry. The matrix material, ice, is assumed to be a nonlinear viscoplastic material. Comparison with data shows excellent agreement for a wide range of initial densities and for large volumetric deformations. Calculations are also made to evaluate grain and neck deformation during compaction. The model can be applied to porous metals and foams, although the constitutive law for the matrix material would have to be altered.

   It seems to me that an application of Brown's neck growth theory, or some derivative of it, would be of keen interest today given the incredible tools of X-ray tomography— something that Brown did not have the luxury of having at the time of publication.

If the authors are determined to extend their theory to anisotropic behavior, I would offer the cautionary note that the degree of complexity probably far exceeds any reward. For instance, if one assumes transverse isotropy (reasonable I think), the limited number of coordinate rotations leads to five stress and strain invariants instead of the normal three. In that spirit, I might suggest the following article is of interest.

A.C. Hansen, D.M. Blackketter, and D.E. Walrath, "An invariant based flow rule for anisotropic plasticity applied to composite materials," **Journal of Applied Mechanics**, Vol. 58, 1991.

The similarities of the referenced work with the present manuscript are striking. Finite element micromechanics of a unit cell are utilized with homogenization to produce a macroscale anisotropic plastic constitutive law for composite materials. The additional invariants play a major role in the flow rule and replace the conventional effective stress measure of classical $J_2$ plasticity theory.

At the end of the day, though, my opinion is that the degree of anisotropy of snow is mild enough such that an anisotropic development is unwarranted for mechanical problems of this nature.

**Minor Editorial Suggestions:**

1. Page 2 line 10  The phrase "*finite elements techniques*" might better read as "*finite element techniques*".

2. Page 2 line 13 Should the word "*bound*" be "*bond*"?

3. Page 3 line 1 In the first sentence, I believe the word "*follow*" might better read as "*follows*".

4. Page 3 Line 2 "*The section 3 presents*" might better read as "*Section 3 presents*". There are other instances of this style as well.

5. Page 3 line 16 Should "*in an homogenization*" read as "*in a homogenization*"?

6. Page 3 line 16 The last word *"introduces"* should be singular.

7. Page 5 line 16 I believe the word "*efficiency*" should be "*efficacy*".

8. Page 10 line 7 The phrase "*initial an final*" should read "*initial and final*".

Respectfully submitted.
Andrew C. Hansen

---

## Short Comment (SC1) · 17 Jan 2017

Dear Authors

As already discussed with one of the authors, I think the values of the Young modulus and stress exponent taken for ice in this manuscript can be questioned.

Schulson and Duval 2009 (Creep and Fracture of Ice, chapter 4, part 4.2.2) give a clear explanation about the difficulties arising when trying to estimate the Young modulus from a "classical" mechanical test, since plasticity is activated very soon. They mention the work done by Gammon et al (1983 for instance), based on acoustic wave propagation as being the only one to give "robust" data of Young modulus for ice. In the

"ice community", it is now accepted that the value of the Young modulus, from Gammon et al work, is 9.33 GPa, very far from the 325 MPa taken here...

This value of about 9 GPa is also the one taken by Theile et al. 2011 (Acta Mat) for instance in order to estimate the Young modulus of snow with FE simulations from microCT images of natural snow.

Another question concerns the value of the stress exponent. The stress exponent at secondary creep in ice is known to be 3, based on a strong experimental work (well summarized in Schulson and Duval 2009). Values close to 5 (or 4.5) were found from experiments that were pushed up to the tertiary creep, when dynamic recrystallization (or micro fracturation) comes into play. I guess that this type of behavior is not relevant for the conditions mentioned here?

Maybe these values do not play a significant role in the main results of the paper.... nevertheless, to use well documented values could maybe enhance the quality of this work?

Sincerely

Maurine Montagnat

―――――――――――――――――――

---

## Author Comment (AC1) · 15 Mar 2017

**Manuscript tc-2016-272 Numerical homogenization of the viscoplastic behavior of snow based on X-ray tomography images.**

Antoine Wautier, Christian Geindreau, Frédéric Flin

March 15, 2017

**Response to Anonymous Referee's comments (RC1)**

We gratefully acknowledge the anonymous reviewer for his comments to clarify some points and improve the quality of our manuscript.

All the reviewers' comments have been taken into account to provide a revised version of our manuscript. The major modifications of the manuscript consist in:

- an enriched introduction containing an improved state of the art and a clearer statement of the objectives and the scope of our manuscript.
- a more detailed assessment of both the elastic and viscous material parameters on the homogenized viscous behavior of snow. A slight change in the postprocessing procedure has been made by introducing a characteristic time.

Based on the review, we understand that the main criticisms made by the reviewer on the content of our article are that:

- our model does not represent all the physical processes at stake at the microscale that modify locally the microstructure
- the parameters of our macroscopic Abouaf's model do not incorporate explicitly microstructure information
- the choice to use FEM instead of DEM is not well explained, as DEM is able to take into account grain boundary sliding and particle rearrangement.

We would like to discuss on these three points and explain more in details the main contributions of our work. In the following, we recall the reviewer's comments followed by our answer.

1. The authors indicate that the macroscopic mechanical behavior of snow strongly depends on its microstructure, which is mainly characterized by its density, topology, external loading (elastic, visco-plastic, brittle-failure) temperature and applied strain-rate. What is missing from the list are the mechanical processes that occur at the microscale (e.g., grain boundary sliding, bond breakage, bond sintering and re-sintering after breakage and particle rearrangement; ignoring metamorphic changes) that are critical to define the evolution of the snow under defined loading (or strain rate) conditions in the RVE. Knowledge of the microstructure is only the starting point to define an evolution function. It is the explicit microscale mechanical processes that determine how snow deformation evolves within the RVE and the model described in the current work does not, and cannot, accurately represent the microscale mechanical processes necessary to develop and RVE evolution function.

**Reply**: We acknowledge the fact that our model does not take into account all the physical phenomena acting at the microscale. In the present work, we propose to determine the 3D macroscopic behavior of snow resulting from the viscous deformation (secondary creep) of the ice skeleton only. Our main objective is to formulate a 3D macroscopic law by performing 3D FEM simulations on real 3D images of snow, in order to underline the microstructure influence. This is a first step towards a more complete modeling of the snow behaviour from 3D images, by introducing more mechanisms and by simulating metamorphism changes.

2. The Abouaf (1985) viscoplastic model used in the current work appears to be primarily porosity dependent and the authors do not provide a clear description of how the homogenized snow microstructure is incorporated into the model. The free parameters of the model include four parameters in two porosity dependent functions, neither of which appear to incorporate microscale mechanical processes. There are continuum scale models specifically developed to incorporate observed microscale measurement that might prove more suitable to this effort, if an appropriate evolution function can be determined.

**Reply**: Thanks to the use of 3D images of different snow types, the particular topology of the snow microstructure is taken into account in the snow mechanical response. Indeed, the shape and size of iso-mechanical dissipation surfaces results from the strong coupling between the microstructure and the non-linear behavior of the ice under consideration. In our case, it appears that the viscous isodissipation curves can be fitted by ellipses in the plane of the two first strain and stress invariants. As a result, an Abouaf model seems to be relevant to account for snow visco-plasticity. In this model, for a given value of the exponent n, all the microstructure information is encapsulated in the two parameters f and c which depends, at the first order, on the porosity only. In the past, different expressions of the material functions  $f(\phi)$  and  $c(\phi)$  have been proposed to describe the densification of granular materials or matrix with spherical voids based on experimental data on metal powders [1, 2, 7], micromechanical modelling (cell model) [6] or numerical simulations on simple microstructures [14]. All these functions, which have been identified in a restricted range of porosity (typically  $\phi < 0.4$ ) on materials with strongly evolving microstructures, point out that the porosity remains, at the first order, the main microstructural parameter influencing their macroscopic viscoplastic behavior. The results obtained in our manuscript are consistent with these observations, however simulations on snow samples with very different microstructures and similar densities (but still isotropic) would help validate this assumption. Finally, let us remark that due to the strong coupling between the microstructure and the non-linear behavior of the ice under consideration, these material functions are valid for a given value of n. In the revised version of the paper, these functions have been identified for 3 values of n (2, 3 and 4.5) (see our response to the comments of M. Montagnat). Probably, micromechanical approaches may lead to more microscopic information on the definition of these functions, but this is not straightfoward at this stage when the porosity is very large.

**3**. The finite element method applied in the paper is not capable of accurately simulating the important micro-scale mechanical processes involved in the RVE deformation (primarily,

grain boundary sliding and particle rearrangement). The only techniques that have demonstrated the ability to represent the micro-scale processes in snow to-date are the discrete element method (DEM) (Johnson and Hopkins, 2005; Hagenmuller et al., 2015) and the deformable DEM (Johnson, 1998). The authors mistakenly indicate that DEM models are only useful for simulating grain breakage and grain rearrangement. With appropriate interaction rules at the grain level the DEM is capable of modeling pretty much any physical process limited only by computational capability, which continue to improve at a rapid rate.

**Reply**: We are aware that DEM is also a powerful tool to bridge the gap between micro and macro scale, and in the revised version of our manuscript the relative pros and cons of DEM and FEM applied on X-ray tomography images are discussed more in details (see the introduction). This analysis governed our choice to use a FEM approach.

- In the FEM method, the complex 3D snow skeleton observed by X-ray microtomography can be meshed without loosing any information on the microstructure and different mechanical behavior of the polycrystalline ice can be considered. In the last decade, most of the studies were dedicated to the elastic behavior of snow [13, 11, 15, 10, 18, 16], possibly up to a brittle failure [8]. Concerning the modeling of more complex snow constitutive behaviors, the proposed approaches mainly focus on the modeling of uniaxial compression tests. For instance, [17] has proposed a beam network model based on 3D images to simulate creep of snow whereas [5] used an elasto-plastic constitutive law for ice in order to determine the failure envelope.
- In the DEM method, the snow skeleton is viewed as an assemblage of ice grains interacting between each other through contact points. As mentioned by the reviewer, this method is well suited to model complex interactions taking place at the interface between snow grains such as elasto-viscoplastic contact deformation, grain sintering and bond breakage or sliding possibly leading to grains rearrangement. This method has been already used on 3D idealized assemblages of ice grains [9] to identify microstructural deformation mechanisms of snow and to simulate creep densification process. However, the application of the DEM directly on 3D images obtained by X-ray tomography is not straightforward, since every ice grain constituting the skeleton must be identified. Recently, DEM simulations taking into account cohesion and friction at the contact between grains have been performed on more realistic 3D assemblages of grains deduced from X-ray tomography. However the shape of each ice grain is approximated by a clump of spheres. Moreover, all these simulations have been performed without taking into account the crystalline orientation of each ice grain. These orientations can play an important role on the viscous deformation mechanisms (secondary creep) at the contact between two grains as it has been recently shown by [4, 3]. The granular structure of the snow (grain shape and crystal orientations) can be determined on real 3D images of snow by X-ray Diffraction Contrast Tomography (DCT) images [12]. However, the application of this technique is not straightforward and very few images are now available.

Both methods applied on X-ray tomography present some advantages and limitations, but we believe that these two approaches can still bring some interesting and complementary results in snow mechanics. Finally, we would like to mention that the proposed methodology (based on isodissipation curves) to identify and formulate the 3D viscoplastic behavior of snow can be also applied by performing DEM simulations or by using a micromechanical approach.

**\**

References

- [1] M Abouaf. Modélisation de la compaction de poudres métalliques frittées. PhD thesis, 1985.
- [2] M. Abouaf and J.L. Chenot. Finite element simulation of hot isostatic pressing of metal powders. Int. J. for Num. Meth. in Eng., 25:191–212, 1988.
- [3] A. Burr, A. Philip, and C.L. Martin. Etude expérimentale de la déformation viscoplastique de cylindres monocristallins de glace. In *Plasticité 2015, Autrans, 28-30 avril*, 2015.
- [4] A. Burr, P. Trecourt, A. Philip, and C.L. Martin. Densification of firm using the discrete element method. In ESMC 2015, 9th European Solid Mechanics Conference, Madrid 6-10 July, 2015.
- [5] Chaman Chandel, Praveen K Srivastava, and P Mahajan. Micromechanical analysis of deformation of snow using X-ray tomography. *Cold Regions Science and Technology*, 101:14–23, 2014.
- [6] J.M. Duva and P. D. Crow. The densification of powders by power-law creep during hot isostatic pressing. Acta Metall Mater., 40:31–35, 1992.
- [7] Christian Geindreau, Didier Bouvard, and Pierre Doremus. Constitutive behaviour of metal powder during hot forming.: Part ii: Unified viscoplastic modelling. *European Journal of Mechanics-A/Solids*, 18(4):597–615, 1999.
- [8] Pascal Hagenmuller, Thiemo C. Theile, and Martin Schneebeli. Numerical simulation of microstructural damage and tensile strength of snow. *Geophys. Res. Lett.*, 41(1):86–89, 2014.
- [9] J. B. Johnson and M. A. Hopkins. Identifying microstructural deformation mechanisms in snow using discrete element modeling. *Journal of glaciology*, 51, 2005.
- [10] Berna Köchle and Martin Schneebeli. Three-dimensional microstructure and numerical calculation of elastic properties of alpine snow with a focus on weak layers. *Journal of Glaciology*, 60(222):705–713, 2014.
- [11] R. A. Pieritz, J.-B. Brzoska, F. Flin, B. Lesaffre, and C. Coléou. From snow X-ray microtomograph raw volume data to micromechanics modeling: first results. Ann. Glaciol., 38:52–58, 2004.
- [12] S. Rolland du Roscoat, S. King, A. Philip, A. Reischig, P. Ludwig, F. Flin, and J. Meysonnier. Analysis of snow microstructure by means of X-ray diffraction contrast tomography. *Advanced Engineering Materials*, 13:128–135, 2011.
- [13] Martin Schneebeli. Numerical simulation of elastic stress in the microstructure of snow. Annals of Glaciology, 38(1):339–342, 2004.

- [14] P. Sofronis and R.M. McMeecking. Creep of power-law material containing spherical voids. Journal of Applied Mechanics, 59:S88–S95, 1992.
- [15] PK Srivastava, P Mahajan, PK Satyawali, and V Kumar. Observation of temperature gradient metamorphism in snow by X-ray computed microtomography: measurement of microstructure parameters and simulation of linear elastic properties. Annals of Glaciology, 51(54):73–82, 2010.
- [16] Praveen K Srivastava, Chaman Chandel, Puneet Mahajan, and Pankaj Pankaj. Prediction of anisotropic elastic properties of snow from its microstructure. *Cold Regions Science and Technology*, 125:85–100, 2016.
- [17] T. Theile, H. Lowe, T. C. Theile, and M. Schneebeli. Simulating creep of snow based on microstructure and anisotropic deformation of ice. Acta Materialia, 59:7104–7113, 2011.
- [18] A Wautier, C Geindreau, and F Flin. Linking snow microstructure to its macroscopic elastic stiffness tensor: A numerical homogenization method and its application to 3D images from X-ray tomography. *Geophysical Research Letters*, 42(19):8031–8041, 2015.

---

## Author Comment (AC2) · 15 Mar 2017

Manuscript tc-2016-272
**Numerical homogenization of the viscoplastic behavior of snow based on X-ray tomography images.**

Antoine Wautier, Christian Geindreau, Frédéric Flin

March 15, 2017

**Response to Anonymous Referee's comments (RC2)**

We gratefully acknowledge the anonymous reviewer for his comments to clarify some points and improve the quality of our manuscript. Based on the reviewer comment, we understand that the aim of our paper is not outlined clearly enough neither in the abstract, the introduction nor the conclusion. We took this remark into consideration and provided a revised version of our paper making this goal easier to understand without any required mechanical background.

All the reviewers's comments have been taken into account to provide a revised version of our manuscript. The major modifications of the manuscript consist in:

- an enriched introduction containing an improved state of the art and a clearer statement of the objectives and the scope of our manuscript.

- a more detailed assessment of both the elastic and viscous material parameters on the homogenized viscous behavior of snow. A slight change in the postprocessing procedure has been made by introducing a characteristic time.

Below, we recall the reviewer's comments and provide some explanations.

**1**. *After reading the abstract, introduction and conclusion, looking at the various figures, and skimming the somewhat "over my head" mathematical development, I am still quite confused about what this paper is about and what it is trying to do. I take it that there is an ultimate goal of providing a simple rule (or demonstration of a rule?) known as "homogenization" that, when applied to snow, allows the treatment of the tortuously complex micro-structure of the snow grain assemblage to be done as if the whole mess were a single homogeneous fluid. I struggled to see where the paper demonstrates either success or failure (and by what criteria?) in achieving this overall goal.*

**Reply**: The reviewer mostly get what "homogenization" is. The idea is to replace a heterogeneous material by an equivalent homogeneous one whose macroscopic behavior take into account the complex heterogeneities existing at the microscale. This procedure is of particular interest when the scale of study (typically the scale of the snowpack in our case) is too big compared to the scale of the heterogeneities (typically here the grain scale) to model these heterogeneities extensively.
Based on the use of snow volumes large enough to be representative of the overall mechanical behavior, our paper provides a systematic method to derive the macroscopic homogeneous

visco-plastic behavior of snow. The main goal of the article is to provide a macroscopic formulation of the snow visco-plastic behavior of snow, valid for any type of isotropic snow. To this respect, our goal is reached and the obtained formulation is summarized in the beginning of section 5. This formulation can be implemented quite easily in finite element codes to compute the densification of the snowpack induced by any kind of mechanical loading and resulting from the secondary creep of the ice skeleton at microscale.

**2**. *I also found the paper to be very loosely tied to a snow phenomena that I am familiar with. Of course, not being a specific snow scientist, but rather a generalist in glaciology, I could be simply too unaware of what is specific to the field. But usually, I get hints from reading the gist of the introduction and abstract as to what the specific phenomena to be illuminated is. I did not get such a hint in this paper.*

**Reply**: The specific phenomena dealt with in this paper is the study of the visco-plastic mechanical behavior of snow resulting from the secondary creep of the ice skeleton at the microscale.

**3**. *Finally, I don't understand why viscoelasticity is being examined. Aren't the two end member rheological treatments (pure viscous for long-term creeping problems, or pure elasticity for short term shock events) good enough for practical snow problems? What is the motivation for developing a viscoelastic treatment? What problem needs it?*

**Reply**: The objective of the paper is to characterize the visco-plastic behavior of snow. However, from a numerical point of view, the computation of the visco-plastic response of ice cannot be achieved without also modeling its elastic behavior. This is the reason why ice is modeled as an elasto-visco-plastic material.

**4**. *The demonstration is also somewhat opaque due to the fact that a black-box software tool (ABAQUS) is used for much of the computation... Some of this computation is non trivial...*

**Reply**: Even if ABAQUS is a commercial software, it is flexible enough to be widely used for research purpose. We chose to use this software because:

- the ABAQUS plugin HomTools developed by researchers at the LMA (acoustic and mechanics laboratory at Marseille, France) enables an easy implementation of the KUBC boundary value problem;

- the constitutive behavior for ice used in this paper is one of the standard constitutive law already implemented in ABAQUS.

**5**. *The paper reads very much like a textbook; and I wonder how much of the mathematical development is simply boilerplate that is also published elsewhere.*

**Reply**: The references for the mathematical development are given in the paper. The mathematical formulations are recalled for the sake of clarity and to justify the homogenization approach carried out in the paper. They are a mean to derive rigorously the 3D macroscopic visco-plastic behavior of snow which is, to our knowledge a new and original contribution to snow mechanics.

---

## Author Comment (AC3) · 15 Mar 2017

Manuscript tc-2016-272
**Numerical homogenization of the viscoplastic behavior of snow based on X-ray tomography images.**

Antoine Wautier, Christian Geindreau, Frédéric Flin

March 15, 2017

**Response to Andrew Hansen's comments (RC3)**

We gratefully acknowledge the reviewer, Andrew Hansen, for his comments to clarify some points and improve the quality of our manuscript.

All the reviewers' comments have been taken into account to provide a revised version of our manuscript. The major modifications of the manuscript consist in:

- an enriched introduction containing an improved state of the art and a clearer statement of the objectives and the scope of our manuscript.

- a more detailed assessment of both the elastic and viscous material parameters on the homogenized viscous behavior of snow. A slight change in the postprocessing procedure has been made by introducing a characteristic time.

**1**. *I was misled by the title. When I first read it, I thought of a finite (large) strain viscoplastic constitutive model for snow where the important deformation mechanisms include such things as bond fracture, intergranular glide, neck growth, etc. That is clearly not the topic of the article. Perhaps the title could be modified to reflect this point by simply adding words such as small strains or some other appropriate descriptor. I'll note that this point is a recurring theme in my review.*

**Reply**: We agree that the domain of application of our formulation of the snow viscoplastic behavior was not obvious. This point has been clarified in the introduction of our revised manuscript. Indeed, as our model is formulated in terms of strain rate, there is no need to specify whether it is valid in small strain or finite strain. However, in order to avoid any influence resulting from the change in density of the samples during the numerical simulations, the simulated time was limited to $40,000$ s corresponding to a volumetric strain of roughly 1.2 % under the considered strain rate. Then, the macroscopic law is generalized to finite deformation problems thanks to the use of a collection of 3D snow images exhibiting different microstructures and densities. At the macroscopic scale, the change in density resulting from large deformations is accounted for by the change in the $f$ and $c$ values. It is clear that it supposes that the density changes are sufficient, at the first order, to capture the influence of the complex modifications of the snow microstructure on its macroscopic behavior. This approximation has been extensively used in the past to described the complete densification (large deformation) of granular materials (metallic powders), in the porosity range $[0 , 0.4]$ [1]. In the future, simulations on snow samples

with very different microstructures but with similar density would help better evaluate this assumption.

**2**. *Page 1 lines 15-18 Echoing the point above, lines 15-18 contain a sentence: In practice, a good knowledge of the macroscopic mechanical behavior of snow in a wide range of applied loads, strain rates, and temperatures is of particular interest with respect to avalanche risk forecasting or to determine the forces on avalanche defense structures.*
*This sentence led to further confusion, for me, in that I again immediately thought of finite deformation (large strain) problems where constitutive modeling is an enormous and important challenge.*
*In reality, this paper is concerned with very small strains, and perhaps small strain rates, where bond fracture and intergranular glide do not occur. More specifically, the original ice microstructure must be intact. I believe this constraint is pretty severe for low density snow where bond fracture will surely initiate at very low strains.*
*I think the reader would be well served if the authors clarify this point by adding a paragraph in the introduction describing the range of applicable strains where they believe the theory is valid. Quantifying such strain levels may be hard to do so an alternate approach would be to simply state that the microstructure is assumed unaltered by bond fracture and further describe the types of problems one might address with the theory. For example, density consolidation under the external body force of gravity seems to me to be the topic of most relevance.*

**Reply**:  We agree with the reviewer's comment and we added some precisions in the revised version of the manuscript to stress the fact that the strain levels considered in the homogenization procedure should remain small in order to avoid any bond fracture and intergranular glide and to keep the volumetric strain small.
As pointed out by the reviewer, the main application of the work carried out lies in a better description of the densification of the snowpack under its own weight.

**3**.  *Page 3 line 3 It would be helpful, at least to me, to provide a brief description of isodissipation curves and how they will be utilized. Are these curves analogous to a yield surface in rate-independent plasticity? This is an area of the manuscript where I was a bit lost and I suspect others will suffer a similar plight.*

**Reply**:  Yes, isodissipation curves can be seen as an equivalent of a yield surface. Indeed, if the yield function is replaced by the mechanical dissipation $\mathcal{P}$, an isodissipation curve corresponding to a mechanical dissipation $\mathcal{P}^\circ$ is described by the implicit equation $\mathcal{P}(\Sigma) - \mathcal{P}^\circ = 0$ in the stress space. Details have been added in the introduction of the revised version of the manuscript.

**4**.  *Page 3 line 30. It would be useful to point out here that* **E** *is the small strain tensor at the macroscale. Again, I was initially confused as in finite strain theory,* **E** *often refers to the Lagrangian finite strain measure. Moreover, finite strain problems are commonplace in snow mechanics.*

**Reply**:  The reviewer is right and the precision was added in the revised manuscript. We also underlined the fact that uppercase letters systematically refer to macroscale quantities whereas lowercase letters are used for their microscale counterparts.

**5**.  *Page 3 lines 16-27. The discussion of the various types of boundary conditions is excellent and I completely agree with the authors. I suspect a formal study of the differences between periodic boundary conditions and kinematically uniform boundary conditions would show very little difference in a homogenization process. The volume averaging process is very forgiving.*

**Reply**:  Thank you for pointing this out! Whatever the boundary conditions used, they introduce errors in the volume averaging process. But they eventually vanish in the volume averaging process as the resulting errors are proportional to the boundary surface.

**6**.  *Page 7 line 3. The focus of this paper is on assumed isotropic behavior (I believe that is a good thing!). Indeed, on line 1 of page 8 this is explicitly stated. Hence, on page 7, line 3 it might be useful to state that, for an isotropic material, the stiffness tensor can be obtained with a single simulation of an REV. Six simulations are sufficient to fully characterize orthotropic behavior.*

**Reply**:  We totally agree with this comment. Considering anisotropic behavior will increase significantly the number of invariants required to formulate the macroscopic behavior. However, before page 8, this assumption is not needed in the theoretical development presented.

**7**.  *Page 7 lines 12-13. I was completely lost by the phrase unit sphere in the second order tensor space.*

**Reply**:  In the vector space composed of the second order tensors, it is possible to define a norm (for instance $||\mathbf{E}|| = \sqrt{\mathbf{E} : \mathbf{E}}$). Because of the homogeneity property written in equation (11), the mechanical response $\mathbf{\Sigma}(\mathbf{E})$ can be deduced from the mechanical response associated with $\mathbf{E}/||\mathbf{E}||$, which is a second order tensor belonging to the unit sphere. Details have been added in the revised version of the manuscript.

**8**.  *Page 9, lines 16-17. I don't believe this sentence is properly expressed. While there is certainly an elastic strain and a viscous strain, there is only one stress. That is, we do not have an elastic stress and a viscous stress. The equation for stress is correct in these lines, the wording is not.*

**Reply**:  We agree with the reviewer and we modified the sentence accordingly.

**9**.  *Page 15 line 8. The ability of snow to dissipate some energy . Perhaps it would be better to say: The ability of snow to dissipate significant energy . A minor point perhaps but a reflection of my background.*

**Reply**:  We agree with the reviewer and we modified the sentence accordingly.

**10**.  *Page 18 lines 11-14. I agree with the components identified in the strain tensor, all zero except $\mathrm{E}_{zz}$. However, I am confused by the stress tensor. Specifically, the authors show $\Sigma_{\theta\theta} = \Sigma_{rr}$. While this may be correct, it is not readily apparent to me. Perhaps the authors could show this point, at least in their review response if not the manuscript.*

**Reply**:   This is due to the mechanical equilibrium $\text{div}(\mathbf{\Sigma}) = 0$. The explanation has been added in the revised version of the manuscript.

**11**.     *Page 20 lines 1-4. Similar comments to the previous point. In the case of stress, it is unclear why $\Sigma_{\theta\theta} = \Sigma_{rr}$. For example in a thick walled pressure vessel $\Sigma_{\theta\theta}$ is vastly different than $\Sigma_{rr}$. Moreover, why is $E_{\theta\theta} = E_{rr}$? My instincts tell me that this problem is over-constrained and that these two conditions involving stress and strain cannot be met simultaneously. Again, perhaps I am wrong but additional clarity is needed.*

**Reply**:   For the stress components, the answer is the same as above while the condition for the strain comes from the constitutive equation (32). The explanation has been added in the revised version of the manuscript.

**12**.     *Page 23 lines 16-18. This comment is important and follows back to my earliest comments about limitations of the theory to very small strains. The authors use the phrase few percents of deformation. I'm guessing even strains of this low magnitude may be too high to preclude bond breaking, particularly at low densities.*

**Reply**:   The homogenization approach proposed in our paper relies upon an incremental approach. Indeed, given a snow sample, the incremental visco-plastic behavior is computed thanks to numerical simulations using very small strain increments (indeed $4.10^{-3}$) while the finite strain problem can be addressed by changing the reference tomography image according to the density evolution. This comes back to our response to your first comment about the title of our manuscript. However, as far as the mechanical of Bartelt and von Moos is concerned, we agree with the reviewer: the microstructure evolutions might exceed the implicit changes in the microstructure taken into account in our model.

**13**.     *Page 24 lines 8-11 I respectfully disagree with the authors as to the importance of this study being extended to anisotropic snow types. In my view, there are so many more important avenues of discovery. I'll offer up a few suggestions of personal interest:*

**Reply**:   We agree with the suggestions made by the reviewers and we added some of them as outlooks in our conclusion.

The minor editorial suggestions were taken into account in the revised version of the manuscript.

*

References

[1] L. Sanchez, E Ouedraogo, L. Federzoni, and P. Stutz. New viscoplastic model to simulate hot isostatic pressing. *Powder metallurgy*, 45:329–334, 2002.

---

## Author Comment (AC4) · 15 Mar 2017

Manuscript tc-2016-272
**Numerical homogenization of the viscoplastic behavior of snow based on X-ray tomography images.**

Antoine Wautier, Christian Geindreau, Frédéric Flin

March 15, 2017

**Response to Maurine Montagnat's comments (SC1)**

We gratefully acknowledge Maurine Montagnat for her comments to improve the quality of our manuscript.

All the reviewers' comments have been taken into account to provide a revised version of our manuscript. The major modifications of the manuscript consist in:

- an enriched introduction containing an improved state of the art and a clearer statement of the objectives and the scope of our manuscript.

- a more detailed assessment of both the elastic and viscous material parameters on the homogenized viscous behavior of snow. A slight change in the postprocessing procedure has been made by introducing a characteristic time.

The specific influence of the material parameters of the ice on their numerical results is discussed below.

**1**. *As already discussed with one of the authors, I think the values of the Young modulus and stress exponent taken for ice in this manuscript can be questioned.*

*Schulson and Duval 2009 (Creep and Fracture of Ice, chapter 4, part 4.2.2) give a clear explanation about the difficulties arising when trying to estimate the Young modulus from a "classical" mechanical test, since plasticity is activated very soon. They mention the work done by Gammon et al (1983 for instance), based on acoustic wave propagation as being the only one to give "robust" data of Young modulus for ice. In the "ice community", it is now accepted that the value of the Young modulus, from Gammon et al work, is 9.33 GPa, very far from the 325 MPa taken here...*

*This value of about 9 GPa is also the one taken by Theile et al. 2011 (Acta Mat) for instance in order to estimate the Young modulus of snow with FE simulations from microCT images of natural snow.*

**Reply**: Even if the value of about 9 GPa seems to be the most widely accepted value for the Young's modulus within the ice community, reported values in the literature cover 2 orders of magnitude from 0.2 GPa to 9.5 GPa as mentioned in [1]. As a result, we chose a value of 325 MPa based on our experimental results. However in the revised version of our manuscript, the value of 9 GPa was also considered. The typical stress response of a snow sample under a constant given strain rate is illustrated on Figure 1 for the snow sample *RG_1600* for the cases $(n, E) = (4.5, 325 \text{ MPa})$ and $(n, E) = (4.5, 9 \text{ GPa})$. The mechanical response is characterized by a transient regime driven by the elastic properties followed by

a permanent regime dominated by the viscoplastic behavior. Because of the change in the Young'modulus value, the duration of the transient regime is changed. In order to compare the time responses of the two aforementioned cases in Figure 1, a characteristic time $\tau$ is defined as the ratio between the ice viscosity $\eta(\dot{E}_{\text{ref}}) = (\dot{E}_{\text{ref}}/A)^{1/n}/\dot{E}_{\text{ref}}$ and the Young modulus $E$

$$\tau \;=\; \frac{\eta(\dot{E}_{\text{ref}})}{E} = \; \frac{1}{E}\left(A^{-\frac{1}{n}}\;\dot{E}_{\text{ref}}^{\frac{1-n}{n}}\right). \tag{1}$$

In Figure 1, the responses $S_1(t/\tau)$ and $\bar{S}_2(t/\tau)$ are found independent of the Young's modulus value chosen. As a result, the homogenization approach presented in our manuscript and, more precisely the macroscopic 3D viscoplastic behavior of snow deduced from ou numerical simluations is completely independent of the Young's modulus value chosen.

The revised version of the paper has been modified accordingly. The second step of postprocessing procedure is slightly modified (in the form) and the initial and final asymptotes are computed with respect to $t/\tau$ and not $t$.

[Figure]

Figure 1: Imposed strain rate (top) and stress response (bottom) versus dimensionless time $(t/\tau)$ for the sample *RG_1600*. Loading strain rate is characterized by $\theta = 65°$ in equation (21). Two Young's moduli and three values of $n$ are considered.

**2**. *Another question concerns the value of the stress exponent. The stress exponent at secondary creep in ice is known to be 3, based on a strong experimental work (well summarized in Schulson and Duval 2009). Values close to 5 (or 4.5) were found from experiments that were pushed up to the tertiary creep, when dynamic recrystallization (or micro fracturation)*

comes into play. I guess that this type of behavior is not relevant for the conditions mentioned here?

Maybe these values do not play a significant role in the main results of the paper... nevertheless, to use well documented values could maybe enhance the quality of this work?

**Reply**: Concerning the visco-plastic parameters used, even if the most commonly used value for the exponent $n$ of the Norton Hoff's law is 3, it is found to vary between 1.8 and 4.6 under usual loading (strain rate, stress) and temperature conditions [3, 5, 4]. The value of 4.5 initially chosen was based on a relaxation test performed on an ice cylinder. In the revised version of the manuscript, two other values of $n$ ($n = 2$ and $n = 3$) are considered to show its influence on the material parameters $f$ and $c$.

As expected, the Figure 1 shows that for a given imposed strain rate and a given sample, the macroscopic responses $S_1(t/\tau)$ and $\bar{S}_2(t/\tau)$ depend on the value of $n$. The viscous stress increases with increasing $n$, since the ice viscosity $\eta(\dot{E}_{\mathrm{ref}})$ increases. Figure 2 shows the influence of $n$ onto the isodissipation curves for the particular snow sample $MF\_522$. Similar results have been obtained on the other samples. As expected, for a given value $\mathcal{P}_{\mathrm{v}}^{\circ}$, the size of the isodissipation curves increases with $n$ (since the ice viscosity $\eta(\dot{E}_{\mathrm{ref}})$ increases) but their shape remains unchanged. They can be deduced from each other by simple dilation.

The values for $f(\phi)$ and $(\phi)$ for $n \in \{2, 3, 4.5\}$ deduced from these simulations are summarized in Table 1 and reported in Figure 3. By contrast to the Young's modulus, the exponent $n$ has a noticeable influence on the parameters $f$ and $c$ and consequently on the parameters $a$, $b$, $p$ and $q$ given in Table 2. The influence of $n$ on these parameters is illustrated in Figure 4.

[Figure]

Figure 2: Influence of the exponent $n$ onto the isodissipation curves ($\mathcal{P}_{\mathrm{v}} = \mathcal{P}_{\mathrm{v}}^{\circ}$) for the particular snow sample $MF\_522$. The associated strain flow vectors $(E_1, \bar{E}_2)$ are represented by solid arrows. Abouaf models are fitted to the numerical points (solid lines) and theoretical values of strain flow vectors are shown (dashed arrows).

Table 1: Optimal values for the parameters $f$ and $c$ of the Abouaf's equivalent stress (25) for three values of $n$.

| Sample name | Porosity | $n=2$ | | $n=3$ | | $n=4.5$ | |
|---|---|---|---|---|---|---|---|
| | | $f$ | $c$ | $f$ | $c$ | $f$ | $c$ |
| PP_123kg_600 | 0.87 | 36.0 | 150 | 79.7 | 336 | 146 | 628 |
| RG_172kg_600 | 0.81 | 16.3 | 75.7 | 33.5 | 156 | 58.3 | 277.4 |
| RG_256kg_512 | 0.72 | 4.05 | 20.5 | 6.98 | 34.7 | 10.5 | 52.3 |
| RG_1600 | 0.64 | 2.07 | 11.0 | 3.32 | 17.0 | 4.70 | 24.0 |
| RG_430kg_651 | 0.53 | 0.915 | 6.38 | 1.40 | 9.12 | 1.89 | 12.1 |
| MF_522kg_542 | 0.43 | 0.354 | 3.32 | 0.503 | 4.26 | 0.630 | 5.07 |

[Figure]

Figure 3: Evolution of the Abouaf coefficients $f$ and $c$ (numerical results as diamond points, functions (29) as solid lines) with respect to the snow compacity for different $n$ values.

All these new results have been incorporated in the revised version of the manuscript. In addition the Figures 6, 8 and 11 of the paper were updated to show the influence of the exponent $n$ on the results. The exponent $n$ doesn't have any influence on the shape of the isodissipation curves but only on the stress level leading to a given isodissipation. Finally, the evolution of the ratio $\Sigma_{rr}/\Sigma_{zz}$ in the case of an oedometer test is similar to the one measured by [2] and [6] on metallic powders. This ratio is almost independent of the exponent $n$, which is consistent with the experimental results of [6].

Table 2: Optimal parameters chosen for the expressions (29) for different $n$ values.

|     | $n = 2$ | $n = 3$ | $n = 4.5$ |
| --- | --- | --- | --- |
| $a$ | 0.68 | 1.0 | 1.5 |
| $p$ | 2.1 | 2.3 | 2.5 |
| $b$ | 4.0 | 6.1 | 8.9 |
| $q$ | 2.0 | 2.2 | 2.3 |

[Figure]

Figure 4: Evolution of the fitted coefficients $a$, $p$, $b$ and $q$ with respect to the exponent $n$ of the ice Norton Hoff constitutive behavior.

\*

References

[1] Chaman Chandel, Praveen K Srivastava, and P Mahajan. Micromechanical analysis of deformation of snow using X-ray tomography. *Cold Regions Science and Technology*, 101:14–23, 2014.

[2] C. Geindreau, D. Bouvard, and P. Doremus. Constitutive behaviour of metal powder during hot forming. part i : Experimental investigation with lead powder as a simulation material. *Eur. J. Mech. A/Solids*, 18:581–596, 1999.

[3] Carlo Scapozza and Perry Bartelt. Triaxial tests on snow at low strain rate. part ii. constitutive behaviour. *Journal of Glaciology*, 49(164):91–101, 2003.

[4] Stefan Schleef, Henning Löwe, and Martin Schneebeli. Hot-pressure sintering of low-density snow analyzed by X-ray microtomography and in situ microcompression. *Acta Materialia*, 71:185–194, 2014.

[5] Erland M Schulson, Paul Duval, et al. *Creep and fracture of ice*. Cambridge University Press Cambridge, 2009.

[6] P. Viot and P. Stutz. Nouveau dispositif expérimental pour l'étude du comportement viscoplastique des poudres métalliques à hautes températures : application à une poudre de cuivre. *C. R. Mecanique.*, 330:653–659, 2002.

---

## Author Response (AR2)

Manuscript tc-2016-272

**Numerical homogenization of the viscoplastic behavior of snow based on X-ray tomography images.**

Antoine Wautier, Christian Geindreau, Frédéric Flin

May 24, 2017

**Response to editor comments**

Dear Editor,

Thank you for having accepted our manuscript. In the final submitted version of our manuscript we slightly improve the grammar and style. The major changes concern the abstract and the introduction.

Please find below a pdf file in which the grammar and style improvements are made visible.

[revised manuscript text omitted]